# Uncovering minimal pathways in melanoma initiation

Hui Xiao [1], Jessica Shiu[2], Chi-Fen Chen [2], Jie Wu [3], Peijie Zhou [4], Sahil S. Telang[2], Rolando Ruiz-Vega[1], Robert A. Edwards [5], Qing Nie [4,6], Arthur D. Lander [1,6] & Anand K. Ganesan [2] ✉

Melanomas are genetically heterogeneous, displaying mitogen-activated protein kinase mutations and homozygous loss of tumor suppressor genes. Mouse models combining such mutations produce fast-growing tumors. In contrast, rare, slow-growing tumors arise in mice combining *Braf* activation with heterozygous loss of *Pten*. Here we show that similar tumors can arise in albino mice bearing only a *Braf* mutation. Incidence kinetics suggest a stochastic event underlies tumorigenesis in tumors that arise with only a *Braf* mutation, yet de novo mutations or structural variants that could explain the incidence of most tumors could not be found. Single-cell transcriptomics of tumors identify a cell type resembling "neural crest-like" cells in human and mouse melanomas. These exist in normal mouse skin, expand upon *Braf* activation, and persist through serial transplantation; analyses of gene expression suggest they serve as precursors of malignant cells. This state may serve as an intermediate on a slow path to malignancy that may provide a diagnostically and therapeutically important source of cellular heterogeneity.

Melanomas exhibit DNA alterations involving multiple oncogenes and tumor suppressor genes[1–3]. However, because melanocytes possess a high background of mutation, it has been difficult to identify a specific sequence of events that inexorably leads to melanoma[4]. Whereas activating *BRAF* mutations are commonly observed driver mutations in melanomas, occurring in 60% of cases[5], the same mutations are also found in nearly all non-malignant growth-arrested melanocytic nevi[6]. This result, together with the observation that expressing activated *Braf* in mouse melanocytes primarily generates nevi, has led to the view that MAP kinase pathway activation (through mutation of *Braf* or, alternatively, *Nras*) is necessary to initiate melanoma but generally not sufficient.

When activated *Braf* (or *Nras*) in mouse melanocytes is combined with complete loss of function mutations of tumor suppressor genes (*Pten*, *p53*, *Ink4a*, *Cdkn2a*), melanomas arise rapidly[7–9]. It has been hypothesized that, without the help of these additional mutations, activation of *Braf* or *Nras* leads to oncogene induced senescence (OIS)

and cell cycle arrest, explaining why only nevi are formed[10,11]. Recent studies cast doubt on this hypothesis and show both that *Braf* activation does not induce melanocyte senescence in vivo[12], and that nevus cells remain competent to proliferate[13]. At present, the mechanism behind growth arrest following *Braf* activation remains unknown but may involve feedback loops normally involved in melanocyte homeostasis[12].

When activated *Braf* is expressed in the melanocytes of mice, in some cases (depending on the expression construct and genetic background[7,9]), rare melanomas do arise, amongst a background of abundant nevi. While it seems plausible that the spontaneous acquisition of additional mutations explains malignant transformation, that hypothesis has not been tested. Indeed, such melanomas are far less studied than either human melanomas, or mouse melanomas produced when multiple mutations (typically three or more) are introduced simultaneously[14]. While mouse melanomas that combine many mutations can be expected to model advanced human disease, by the

[1]Center for Complex Biological Systems, University of California, Irvine, USA. [2]Department of Dermatology, University of California, Irvine, USA. [3]Department of Biological Chemistry, University of California, Irvine, USA. [4]Department of Mathematics, University of California, Irvine, USA. [5]Department of Pathology, University of California, Irvine, USA. [6]Department of Developmental and Cell Biology, University of California, Irvine, USA. ✉e-mail: aganesan@uci.edu

same token, they may not be a good model for elucidating the stepwise progression that occurs between normal melanocytes and melanoma.

Recent studies suggest that such progression is likely to involve not only the accumulation of mutations but also transitions among cell states. Melanomas display a high degree of intratumoral heterogeneity, both in cellular behaviors and epigenetic modification[15], and stochastic switching between melanoma cell states has been directly observed in vitro[16–20] and proposed to occur in vivo. In some cases, it has been argued that specific states serve as necessary waystations on the path to therapeutic resistance[21,22]. Little is known, however, about what drives transitions between states, nor the extent to which mutational or other processes influence them.

With the advent of methods for assessing transcriptomes at the single cell level, several groups have attempted to define the landscape of cellular heterogeneity in melanoma. Based on analyses of human melanomas, mouse models, and cell lines, gene expression signatures have been proposed to define cell states that have been variously categorized as pigmented, invasive, proliferative, neural crest (NC)-like, mesenchymal, intermediate, and *BRAF* inhibitor resistant[19,22,23]. Signatures derived in different studies are often divergent, and it is unclear whether this reflects real biological differences or variations in data quality or informatic approaches. A complicating factor is that most data come from the analysis of melanomas bearing different combinations of driver mutations (including in some cases ones that were not fully characterized) in a variety of genetic backgrounds.

Here we clarify the cellular complexity of melanoma during early stages of tumor development by focusing on two mouse models of "minimal" induction in which melanocytes are engineered to express only activated *Braf*, or activated *Braf* plus a single null allele of the tumor suppressor *Pten*. On a genetic background that allowed for rare melanoma emergence even when only *Braf* was mutated, the kinetics of tumor appearance were consistent with a requirement for a single stochastic event. We found little evidence that this event involved a consistent DNA mutation or common copy number variation, although we could not rule out the possibility that multiple different mutations might exist that have similar effects in different tumors, or that certain kinds of mutations were missed. Analysis by single cell RNA sequencing of tumors, as well as adjacent and normal skin, revealed that the major cell types in minimally-mutated melanomas expressed relatively low levels of pigmentation genes, and the resulting tumors were indeed hypomelanotic. Through direct comparison of gene expression signatures, it was possible to identify major cell states in such tumors that was similar to a state specifically thought to play a role in the development of therapy resistance[22]. Intriguingly, one of the states we observed was also detected in tumor-free skin, suggesting it is not obligatorily malignant when it is present alone. Using informatic tools we predict that this cell may act as a direct source of more abundant types of tumor cells. We discuss the possible relationship of cell states we identify here to events in human melanoma initiation and progression.

## Results

### *Braf* activating mutations induce rare melanomas in both *Pten*-heterozygous and wild-type albino mice

We used a *Tyrosinase::CreERT2; Braf$^{CA-fl/+}$* construct[24] in transgenic mice to specifically activate *Braf* in melanocytes by topically applying tamoxifen at postnatal days 2-4 (Fig. 1a, Table 1). On a C57BL/6 background, such mice developed abundant nevi in a matter of weeks, but no melanomas. In contrast, when a single floxed allele of *Pten* was introduced into this background, creating a heterozygous *Pten* mutation in melanocytes, sporadic tumors arose, an average of three per animal (Fig. 1a, S1A, B, Table 1). Both gross (Fig. 1a, S1A) and histologic (Fig. 1c) analysis of these tumors strongly suggested they did not arise by loss of heterozygosity of the wild type *Pten* allele. For example, whereas melanomas generated by activating *Braf* together with loss of

both *Pten* alleles (*Pten$^{\Delta/\Delta}$*), are strongly pigmented[7,25] (Fig. 1c, right lower image), these tumors display very little pigment either visually (Fig. 1a, S1A) or histologically (Fig. 1c). In addition, immunofluorescence staining of *Pten$^{\Delta/+}$* tumors confirmed that they expressed Pten protein (Fig. S2A).

We also examined the outcome of expressing *Braf$^{CA/+}$* in the melanocytes of albino mice (congenic with C57BL/6, Fig. S1C). We found that, on an albino background (*Tyr$^{c-2J}$*), *Braf$^{CA/+}$* alone produced melanoma in about 2/3 of animals, with no animals displaying more than one tumor (Fig. 1a, S1A, B). Combining *Braf$^{CA/+}$* with *Pten* heterozygosity in albino mice increased the numbers of tumors per animal to a level similar to that seen in black *Braf$^{CA/+}$ Pten$^{\Delta/+}$* mice (Fig. S1A-B, Table 1). Albino *Braf$^{CA/+}$* tumors also displayed immunologically detectable Pten (Fig. S2A, Table 1).

The rarity of tumors when *Braf* is activated either alone or together with a single loss of function allele of *Pten* suggests at least one additional event is required to produce melanoma. The distribution of times at which tumors become detectable in such mice strongly suggests that the event is statistically random, i.e. occurs with a constant, small probability. As shown in Fig. 1b, plots of tumor-free survival over time are fit by single declining exponential functions, consistent with "single-hit kinetics", i.e., the expectation of a process with a constant probability per unit of time. Because of the need for tumors to exceed a threshold size before being detected, we cannot rule out the possibility of *more* than one random event being required for tumor initiation, but based on the application of the single-hit model, we may extract from the final slopes of logarithmic plots a "half-time" ($t_{1/2}$) for tumor initiation (the expected time for tumors to arise in 50% of animals) equal to ~33 weeks in albino *Braf$^{CA/+}$* mice, ~15 weeks in albino *Braf$^{CA/+}$ Pten$^{\Delta/+}$* mice, and ~6 weeks in black *Braf$^{CA/+}$ Pten$^{\Delta/+}$* mice. It is important to note that, on such plots, the rate of tumor growth is reflected only in the initial lag-time and not the eventual slope, which is a measure of when initiating events occur. Nonetheless, it is clear that tyrosinase function and *Pten* gene dosage both influence the probability of melanoma initiation in *Braf$^{CA/+}$* and *Braf$^{CA/+}$ Pten$^{\Delta/+}$* mice.

To investigate whether the random, initiating event required to produce albino *Braf$^{CA/+}$* tumors is a shared de novo DNA mutation, we performed whole genome sequencing (WGS) on three such tumors (Fig. 1d, e, Table 1). Tumors were dissected from the overlying skin, and DNA was prepared along with control DNA from spleen and skin of the same animals. Sequencing was performed to an average of 80X coverage (within mappable regions, >99% of nucleotides were covered at least 40 fold). A Bayesian somatic genotyping model (Mutect2) was used to identify tumor-specific somatic variants as those present in tumor samples but absent from the spleen of the same animal. Based on the levels of unrecombined and recombined *Braf$^{CA/+}$* alleles detected in each sample, we estimated (see Methods) that the fraction of DNA that was tumor-derived in each sample was 19%, 55%, and 36%, for tumors 1, 2, and 3, respectively, in Fig. 1d,e. As a heterozygous mutation required for malignant transformation should have been observed at half these allele frequencies, the probability, even assuming only 40X coverage, of failing to observe a mutation in one of the three tumors would have been 2%, 0.00025%, and 0.036% respectively.

Overall, we observed between 4000 and 14,100 mutations per sample, including single nucleotide variants (SNVs), insertions, deletions, substitutions, and stop gains. Many were present at low frequency (compared with the expected proportion of tumor genomes in each sample; Supplementary Data 1, 3, 5), very few were exonic, and only a handful of those were nonsynonymous (Fig. 1d; Supplementary Data 2, 4, 6). Of these, no gene was mutated across all three tumors. We noted two genes in which mutations were seen in 2/3 tumors sequenced (Supplementary Data 7). One of these genes, *Pira6*, a gene that is involved in osteoclast function[26], was not expressed in the tumor (no reads in *Pira6* gene were detected in any of the cells in the tumor single cell RNAseq dataset, discussed below) and was thus not

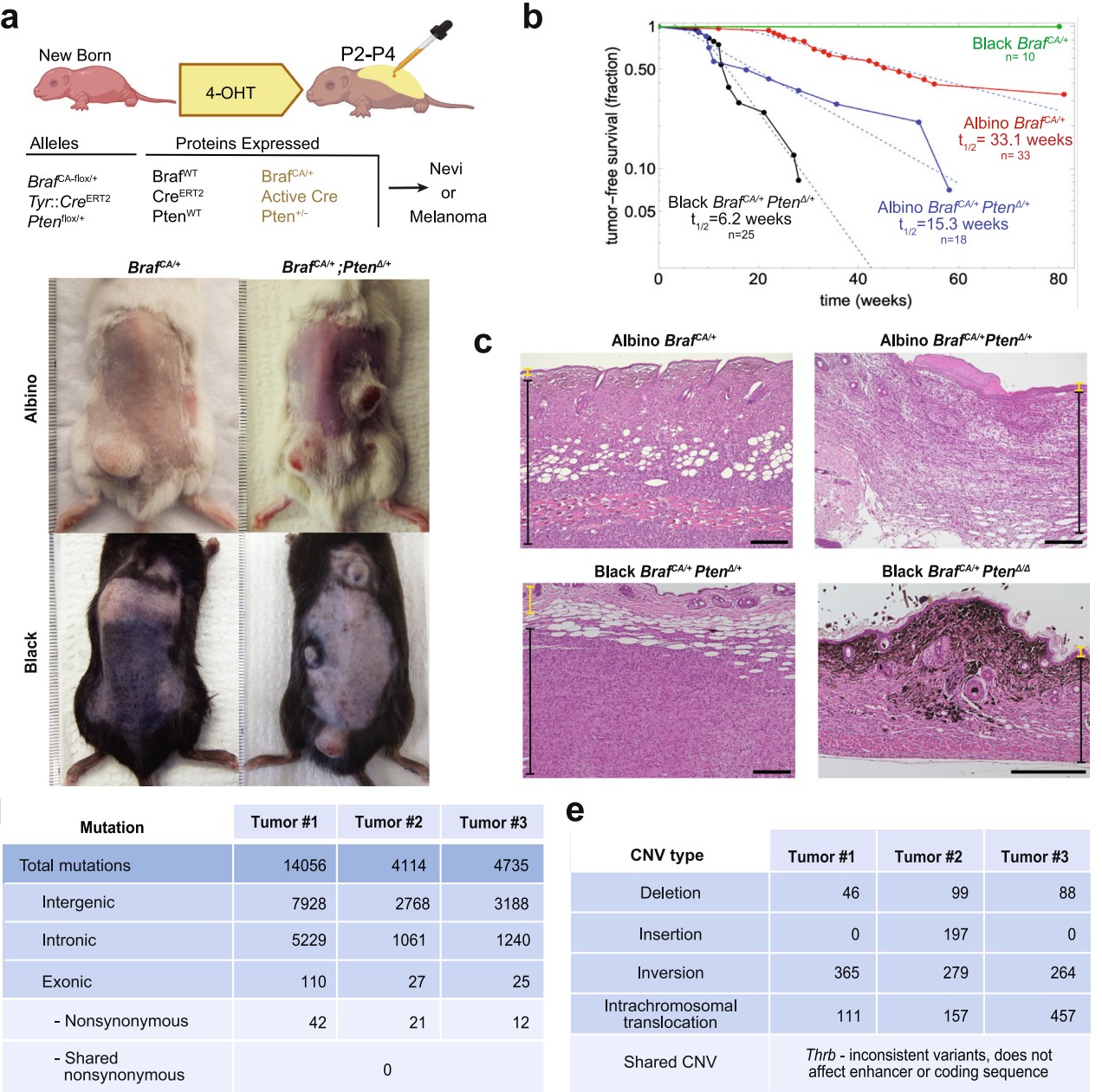

**Fig. 1 | The *Braf*^CA/+ mutation can induce melanomas in mice independently of additional mutations. a** Images and (**b**) tumor-free survival curves for *Braf*^CA/+ and *Braf*^CA/+; *Pten*^Δ/+ mice in different coat-color backgrounds. Albino *Braf*^CA/+ mice develop tumors not observed in congenic black *Braf*^CA/+ mice. Illustration in (**a**) was created in BioRender. Shiu, J. (2025) https://BioRender.com/rhuf7aj. Source data are provided as a Source Data file. **c** Hematoxylin and eosin staining of tumors (representative micrographs of three independent biological replicates are shown). Yellow and black brackets on the side denote tumor-free and tumor-containing regions, respectively. Note that different tumor genotypes differ markedly in pigmentation. *Pten*^Δ/Δ tumors are highly pigmented whereas *Pten*^Δ/+ tumors largely lack pigment. Scale bar: 200 um. **d** Whole-genome sequencing (WGS) was performed on three albino *Braf*^CA/+ tumors (80X coverage). Observed mutations (intergenic, intronic, exonic, and non-synonymous), including single nucleotide variant (SNV), insertion, deletion, substitution, and stop gain are reported. No non-synonymous mutations were shared amongst the tumors. **e** The nearest gene bodies to detected copy number variants in albino *Braf*^CA/+ tumors were determined. Only one gene, *Thrb*, has CNVs close to the gene body in all three tumors. The locations of these CNVs did not suggest functional significance and did not overlap (see Fig. S1D).

investigated further. A second gene, *Flna*, had a mutation near exon 21 in 1/3 tumors and a second mutation in exon 45 in 1/3 tumors. Given the incidence of *Flna* mutations, we further explored the prevalence of *Flna* mutations using our single cell RNAseq data (see below for description of our dataset). We observed Exon 21 *Flna* mutations in 2/7 tumors subjected to scRNAseq (Supplementary Data 8). We observed no Exon 45 *Flna* mutations in tumors in our single cell RNAseq dataset (Supplementary Data 8). Published studies identified the region around exon 21 of *Flna* as a mutation hotspot in mouse melanoma

tumors, with 20-33% of tumors that arose in the absence of UV having a *Flna* mutation around exon 21 regardless of initiating mutation[27,28]. This is consistent with our results here, where we observed *Flna* mutation in 3/10 tumors. According to the three-dimensional structure of the Flna protein[29], the *Flna* Exon 45 mutation removes three amino acids in a flexible loop between two beta-strands in one of the filamin repeats. Whether this would have functional consequences for that domain is difficult to say without direct structure or function studies, but this site was not identified as a hotspot in the other mouse

## Table 1 | Key Resource Table

| Resource type | Designation | Source or reference | Identifiers | Additional information |
|---|---|---|---|---|
| Gene (Mus musculus) | *Braf* | Mouse Genome Informatics (MGI) | MGI:88190 | |
| Gene (Mus musculus) | *Pten* | MGI | | |
| Gene (Mus musculus) | *Cre* | MGI | MGI:3641203 | MGI Transgene name: GeneTg(Tyr-cre/ERT2)13Bos |
| Mouse strain | C57BL/6 J | Jackson Laboratory | 000664 | |
| Mouse strain | Albino B6(Cg)-*Tyr^c-2J*/J | Jackson Laboratory | 000058 | |
| Mouse strain | NSG | Jackson Laboratory | 005557 | |
| Mouse strain | *Sox2^tm1.1Lan*/J | Jackson Laboratory | 013093 | |
| Sequence-based reagent | Braf_F | IDT | PCR primer | 5'-TGAGTATTTTTGTGGCAACTGC –3' |
| Sequence-based reagent | Braf_R | IDT | PCR primer | 5'-CTCTGCTGGGAAAGCGCC –3' |
| Sequence-based reagent | Pten_F | IDT | PCR primer | 5' AAAAGTTCCCCTGCTGATGATTTG T 3' |
| Sequence-based reagent | Pten_R | IDT | PCR primer | 5' TGTTTTTGACCAATTAAAGTAGGCTGT G 3' |
| Sequence-based reagent | Cre_WT_F | IDT | PCR primer | 5'- TTC CCA CAC TTA ACA GCC CCA –3 |
| Sequence-based reagent | Cre_WT_R | IDT | PCR primer | 5'- GGA CGT GTG GAG GGA TCG T-3' |
| Sequence-based reagent | Cre_Tg_F | IDT | PCR primer | 5'- TTC CCA CAC TTA ACA GCC CCA-3' |
| Sequence-based reagent | Cre_Tg_R | IDT | PCR primer | 5'- CCC ACA TCA GGC ACA TGA GT –3' |
| Chemical compound, drug | 4-hydroxytamoxifen | Sigma-Aldrich | 68047-06-3 | |
| Primary Antibody | Aqp1 (1:200-400) | EMD Millipore | AB2219 | Used for immunohistochemistry |
| Primary Antibody | Sox2 (1:500) | Abcam | Ab97959 | Used for immunohistochemistry |
| Primary Antibody | Pten (1:100) | Cell Signaling | 9559S | Used for immunofluorescence |
| Secondary Antibody | Alexa Fluor 594 goat anti-Rabbit IgG (1:1000) | Life Technologies | A11037 | Used for immunofluorescence |
| RNAscope probe | Aqp1 | ACD Bio | 504741-C2 | Used for RNAscope FISH |
| RNAscope probe | Sox2 | ACD Bio | 401041-C3 | Used for RNAscope FISH |
| RNAscope fluorophores | Opal-620 | Akoya Biosciences | FP1495001KT | Used for RNAscope FISH Channel 2 |
| RNAscope fluorophore | Opal-650 | Akoya Biosciences | FP1496001KT | Used for RNAscope FISH Channel 3 |
| Software | Cell Ranger | 10X genomics | RRID:SCR_017344 | |
| Software | Seurat | Stuart et al., 2019 | RRID:SCR_016341 | |
| Software | Harmony | Korsunsky et al., 2019 | RRID:SCR_022206 | |
| Software | scVelo | Bergen et al., 2020 | RRID:SCR_018168 | |
| Software | MuTrans | Zhou et al., 2021 | | https://github.com/cliffzhou92/MuTrans-release |
| Software | BreakDancer | Tattini et al., 2015 | RRID:SCR_001799 | |
| Software | Trimmomatic | Bolger et al., 2014 | RRID:SCR_011848 | |
| Software | FastQC | Andrews et al., 2010 | RRID:SCR_014583 | |
| Software | InferCNV | Patel et al., 2014 | RRID:SCR_021140 | https://www.bioconductor.org/packages/release/bioc/html/infercnv.html |
| Software | SigProfiler | Khandekar et al., 2023 | RRID: SCR_023122 | https://github.com/AlexandrovLab/SigProfilerMatrixGenerator |
| Software | CNVkit | Talevich et al., 2016 | RRID:SCR_021917 | https://github.com/etal/cnvkit |
| Software | Annovar | Wang et al., 2010 | RRID:SCR_012821 | http://www.openbioinformatics.org/annovar/ |

Summary of key tools used in this study, including mouse strains, primers, antibodies, staining probes, software, and pipeline.

melanoma studies[27,28]. We specifically checked, and did not find, mutations in sequences upstream of *Tert* that correspond to sites of promoter mutations in human melanoma[30]. Mutational signatures were also examined using SigProfiler[31] (Fig. S2B). Interestingly, C to T mutation changes were more frequent than other changes even though these mice were not intentionally exposed to UV light (the frequency of these changes was lower than typically observed after UV treatment[27]).

Since the above methods do not detect DNA changes on a larger scale than the sizes of the sequenced fragments, we also used paired-

end whole genome sequencing to identify breakpoints and infer copy number variations (CNVs) and structural variations (SVs). We used the *BreakDancer*[32] pipeline to identify CNVs that were unique to the tumor and not observed in the spleen or skin of the same animal (allowing for a false discovery rate of 0.05, a relatively lenient cutoff). We grouped CNVs by the gene bodies to which they were closest. Only one gene, *Thrb*, had CNVs near the gene body in all three tumors sequenced (Fig. 1e, Supplementary Data 9, 12, 15), and the three observed mutations were all small, intronic, and did not overlap, and thus seemed unlikely to have an impact on gene expression (Fig. S1D). We also did not observe expression of *Thrb* in tumor cells by single cell gene expression (Fig. S4D), suggesting it would be unlikely for a mutation in this gene to contribute to tumor initiation. Of note, we did not observe CNVs in genes that have been identified to display copy number variations in human *Braf* mutant tumors, including tumor suppressor genes *Pten, Cdkn2a, Tert, Nf1, Tp53, Rac1* and *Cdk4*[3].

We also used *CNVkit*[33] to search for potential CNVs in the same tumor samples. CNVkit predicts deletions and duplications from contiguous regions of difference in coverage (read depth), relative to a reference sample (in this case control spleen DNA), adjusted for tumor cell frequency. Because read-depth based algorithms have high false positive rates[34], we used spleen-spleen comparisons to estimate rates of false discovery, and identified generous parameters expected to limit false discoveries to a few hundred per tumor sample (see Methods). CNVs were identified in all tumors but the involved genes overlapped poorly with each other, and even less well with those identified in any tumor by BreakDancer (Supplementary Data 11, 14, 17). No gene known to display copy number alteration in human *Braf* mutant tumors was identified among CNV calls that were shared across tumors. In four cases BreakDancer detected a SNV near those calls in a single tumor, but the breakpoints were inconsistent with the CNV changes suggested by CNVkit, which were therefore judged likely to be false positives.

To better depict the level of structural variants and copy number variants, we plotted the observed structural variants identified by BreakDancer on CIRCOS plots (Fig. S3A) and copy number variants identified by CNVkit on scatterplots (Fig. S3B), as described[33]. When comparing CIRCOS plots depicting structural variants present in skin (after spleen was substracted) and tumors (after skin and spleen were subtracted), we observed that the level of detected structural variants was similar and most of the structural variants identified did not overlap (Supplementary Data 10, 13, 16). In fact, the total number of structural variants in the tumor were 1–3% greater than the number of variants identified in the spleen (Supplementary Data 10, 13, 16), indicating that the number of structural variants in tumors was not significantly greater than the degree of background structural variants observed in skin. Similarly, scatterplots of segment copy number inferences failed to identify any copy number segment alterations conserved across tumors (Fig. S3B).

Finally, we used a third method to search for CNVs in albino *Braf* mutant tumors, which involved searching in single cell transcriptomic data for contiguous blocks of gene expression displaying greater or lower than expected transcript numbers (these data are discussed below in the next section on single cell RNA sequencing). Even when adjusted to a relatively high false discovery rate, this method also failed to identify shared or overlapping structural variants among tumors.

## Identifying melanocytic cell states

The possibility that a non-genetic transition—e.g. an epigenetic cell state change or a collective cell transition—may drive melanoma initiation, at least in albino *Braf*^CA/+ animals, is interesting given that non-genetic, stochastic switching between cell types has been described among melanoma cells in vitro[20]. If such switching plays a role in tumor initiation in *Braf*^CA/+ and *Braf*^CA/+ *Pten*^Δ/+ mouse models, we

reasoned that evidence of it might be found by analyzing the cell types or states within tumors. For example, we might observe a cell state in melanoma tumors that was also found in skin in which tumors had not yet arisen.

To explore this possibility, we performed single cell RNA sequencing on a total of 47 skin and tumor samples from 36 animals: 15 wildtype skin samples, 10 black *Braf*^CA/+ skin samples (which contain only nevi), 3 albino *Braf*^CA/+ tumors, 4 albino *Braf*^CA/+ *Pten*^Δ/+ tumors, 4 black *Braf*^CA/+ *Pten*^Δ/+ tumors, and 11 sample-matched tumor-adjacent-skin samples collected alongside each of the 11 tumor samples (Fig. 2a, Supplementary Data 18, Table 1). After combining samples using the merge function of Seurat, performing quality control, and normalizing using scTransform, a total of 345,427 cells were identified (Table 1). Using unsupervised clustering, based on highly variable genes and marker genes known to identify cell types, we identified 18 cell types, including melanocytes (*Dct*+, *Pmel*+, *Mitf*+, and *Mlana*+), Schwann cells (*Mpz*+, *Dhh*+), fibroblasts (*Pdgfra*+, *Col3a1*+, *Sparc*+) and macrophages (*Adgre1*+, *Cd74*+, *Ctss*+) (Fig. 2b, Fig. S4A, B, Supplementary Data 19). Specifically within tumor samples, we observed an abundant cell type that expressed markers commonly seen in melanoma, including the markers *Plp1, Gpm6b, Postn, Mcam, S100b*[35] (Fig. 2b, d).

We then subsetted the cells in all samples that were judged to have a potential lineage relationship to melanocytes. These included melanocytes themselves, Schwann cells (thought to share a common precursor with melanocytes[36]), and any cell types in which we could document recombination driven by the *Tyr-CreERT2* transgene (in a subset of samples, the mTmG reporter system[37] was used so that expression of GFP marked cells in which Cre-mediated recombination had occurred); the latter included the abundant cell type in tumor samples that expressed melanoma markers.

These 35,527 cells, which we refer to as "NC-derived", were re-normalized using ScTransform and subclustered (Fig. 2c, Fig. S4C, D, Supplementary Data 20). We identified within this group melanocytes, which produce a high level of pigmentation genes (*Ptgds, Pmel, Mlana*) consistent with the highly pigmented nature of nevus melanocytes[12]; a Schwann cell population (marked by *Mpz*); and several tumor-enriched cell populations that clustered separately from melanocytes, and expressed melanoma markers (*Plp1, Gpm6b, Postn, Mcam*[35]) (Fig. 2c, d). A distinct population of cells was observed primarily, but not exclusively, in tumor samples and could be distinguished by weak but detectable expression of pigmentation genes (*Mlana, Tryp1*), expression of multiple "neural" genes (*Bche, Sema3b*), and expression of genes encoding structural components of the extracellular matrix. We termed these LNM (Lightly pigmented, Neural, extracellular Matrix) cells. Other markers of LNM cells included *Aqp1, Dhh*, and *Igf1* (Fig. 2d, Fig. S4D). Although LNM cells were relatively abundant in tumor samples, they were also observed in normal skin, and their numbers increased substantially in the tumor-free skin of *Braf*^CA/+ mice (Fig. S4E). We calculated the Euclidean distance in principal component space between LNM cells, the principal tumor cluster, Schwann cells and melanocytes and observed that the LNM cells were closer to principal tumor cells than to the Schwann cells or melanocytes (Fig. S4F, Supplementary Data 22). Immunohistochemical staining for Aqp1 indicated that LNM cells were sparse in normal skin, located in or around nevi in black *Braf*^CA/+ skin, and were observed throughout the dermis in *Braf*^CA/+ tumors (Fig. S4G, Table 1).

Principal tumor cells, defined by the expression of known melanoma marker genes *Postn*+, *Mcam*+, *Plp1*+, *Gpm6b*+[35], were observed in tumors from all three genotypes (Fig. 2e). These cells could be subclustered into four distinct subsets. Two included cells of all three genotypes, which we refer to as "shared" and "proliferative". The remaining two corresponded to groups unique to either *Pten*^Δ/+ or *Pten*^+/+ tumors (the latter necessarily being albino) (Fig. 2c, d, S4D). The shared cluster was marked by expression of extracellular matrix

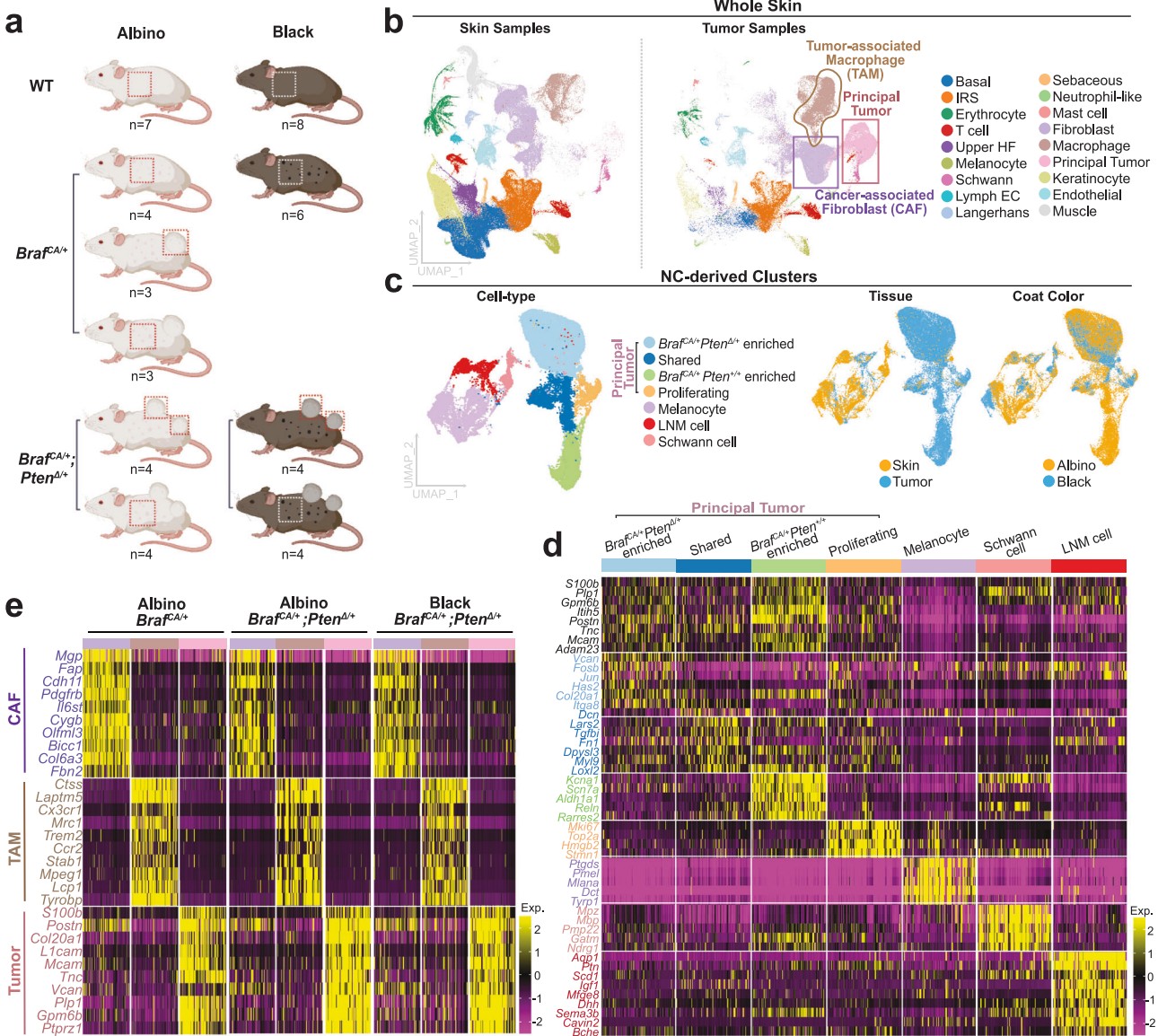

**Fig. 2 | Single cell transcriptomics identifies melanocyte/neural crest-derived, macrophage, and fibroblast populations in normal skin and *Braf*<sup>CA/+</sup> and *Braf*<sup>CA/+</sup>; *Pten*<sup>Δ/+</sup> tumors. a** Coat colors, genotypes and numbers of mice subjected to single-cell RNA sequencing. Created in BioRender. Shiu, J. (2025) https://BioRender.com/2832vxc. **b** 345,427 cells from 36 mice (47 normal, nevus-containing skin, and melanoma samples) from the genotypes in panel A were subjected to scRNAseq, jointly clustered, and projected onto a common UMAP. ScRNA-seq identified populations enriched in tumors, highlighted in the boxed areas. IRS: Inner root sheath. HF: Hair follicle. EC: Endothelial cell. **c** NC-derived clusters (identified by the expression of *S100b*, *Dct*, *Mpz* [as well as a *Cre*-reporter gene in selected cases]) were further subsetted (35,527 cells in total) and populations specific to tissue and coat color were characterized. **d** Differentially expressed genes associated with NC-derived clusters defined in (**c**). Known melanoma markers are labeled in black; colored bars and other text colors correspond to labeling in (**c**). **e** Gene expression profiles of tumor-enriched fibroblasts (purple), macrophages (brown) and tumor cells (pink) compared among the tumor genotypes. Each column in the heatmap represents a cell from the corresponding group.

related genes (*Vcan*, *Col20a1*) and low expression of pigmentation genes (*Mitf*, *Dct*, *Pmel*)[38] (Fig. 2d, e) consistent with the observation that pigment in these tumors was not readily apparent visually or histologically, even in black mice (Fig. 1a, c, S1A). The proliferative cluster was marked by the expression of proliferation markers *Top2a*[39], *Mki67*[39], and *Hmgb2*[40] (Fig. 2c, d, S4D). The *Pten*<sup>Δ/+</sup>-specific cluster was distinguished by expression of genes associated with cell stress (*Fos*, *Jun*)[41], whereas the albino *Pten*<sup>+/+</sup>-specific cluster was marked by expression of genes that have been suggested to mark "neural crest stem cells" (*Aldh1a1*, *Ngfr*, *Reln*)[19,42] (Fig. 2d, Fig. S4D).

We also identified and subclustered fibroblastic and immune cells in both skin and tumor samples. Cancer-associated fibroblasts (CAFs) and tumor-associated macrophages (TAMs) were specific to tumor tissue (Fig. 2b); their gene expression profiles were consistent across tumor genotype (*Braf*<sup>CA/+</sup> *Pten*<sup>+/+</sup> vs. *Braf*<sup>CA/+</sup> *Pten*<sup>Δ/+</sup>) (Fig. 2e). CAFs expressed markers that had been previously identified in other cancer types (*Mgp*, *Col1a1*, *Pdgfrb*, *Pdgfa*)[43] (Fig. S5A). TAMs shared the expression of some macrophage specific markers with tissue macrophages, but failed to express other markers characteristic of tissue macrophages (*Cxcl2*[44] and *Retnla*[45]) (Fig. S5B).

The ability to isolate cells of different types using scRNAseq provided an additional opportunity to look for DNA structural changes specific to tumor cells. In this case, inferCNV (https://github.com/broadinstitute/inferCNV, Table 1) was used to search for large, contiguous blocks of genes that collectively display higher or lower than expected levels of gene expression in principal tumor cells versus

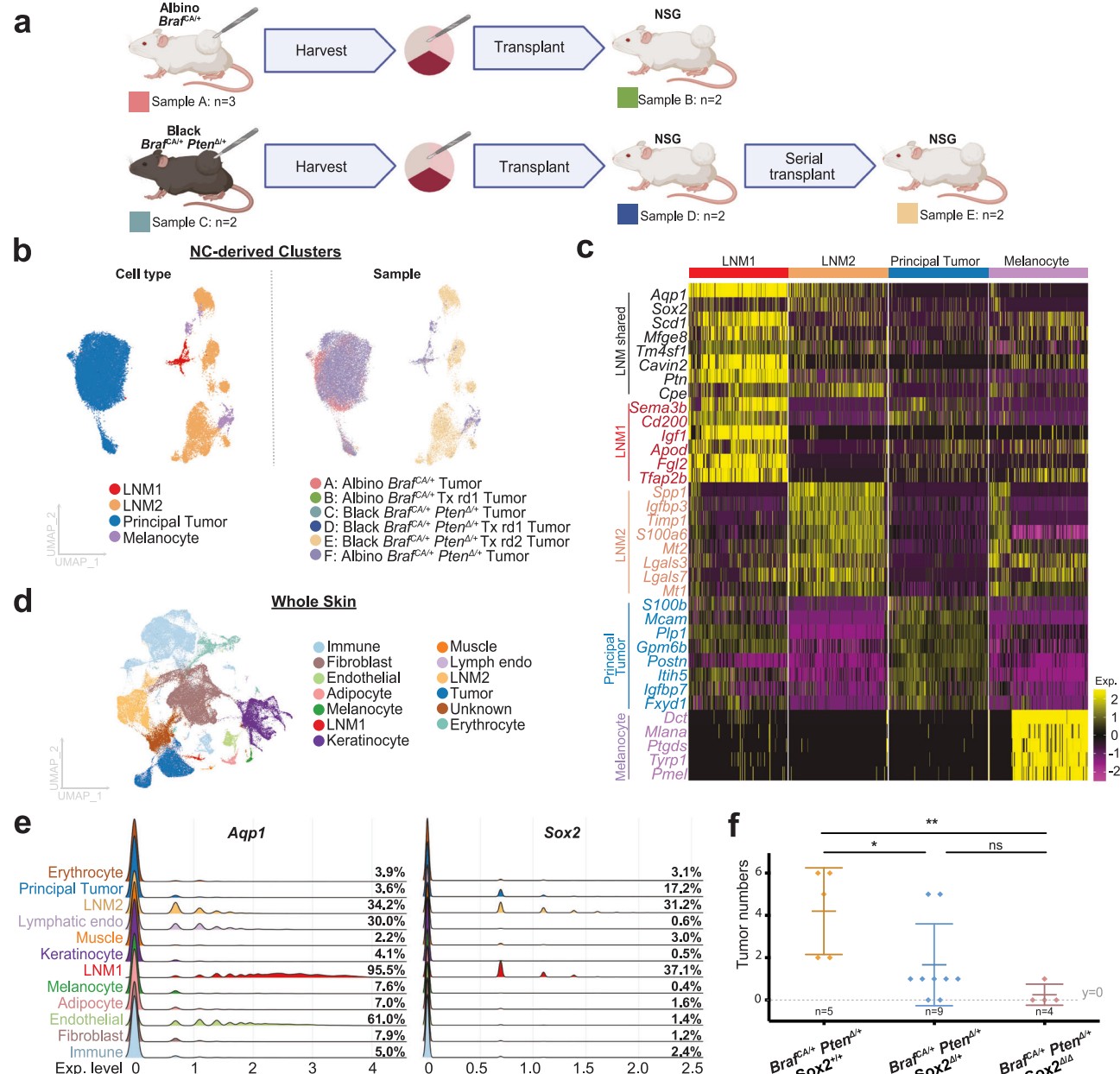

**Fig. 3 | LNM cells persist after transplantation. a** Tumors derived from *Braf*^CA/+ and *Braf*^CA/+; *Pten*^Δ/+ mouse models were passaged by transplantation onto NSG (immune deficient) mice. Created in BioRender. Shiu, J. (2025) https://BioRender.com/m4kraw3. **b** Single-cell gene expression profiles were used to subcluster NC-derived cell clusters (32,611 cells), among which principal tumor, melanocyte and LNM clusters were identified. UMAP visualizations of cell-types and sample distribution are shown. rd = Round of transplantation. Tx = transplant. **c** Gene expression profiles of the NC-derived clusters identified in (**b**). Gene signatures shared among LNM1 and LNM2 are labeled in black. Each column in the heatmap represents a cell from the corresponding group. **d** 182,950 cells from *Braf*^CA/+ and *Braf*^CA/+; *Pten*^Δ/+ primary tumors (n = 9) and transplanted NSG tumors (n = 6) were subjected to single-cell RNA-seq. UMAP visualization of cell-types is shown. **e** Ridge plots of *Aqp1* and *Sox2* expression. Note the high expression of *Aqp1* and *Sox2* in both LNM1 and LNM2, compared to other cell types present in the whole skin. Percentages on the side denote the fraction of cells expressing *Aqp1* and *Sox2* in each cell type, respectively. Exp.: Expression. **f** Conditional *Sox2* deletion in melanocytic/NC-derived cells inhibits tumor development. Tumor incidence in mice of the indicated genotypes is shown, and each dot represents an individual biological replicate; error bars represent the group mean ± SD. Tumor numbers were compared using an unpaired two-tailed *t*-test. Asterisks denote statistical significance: $p < 0.05$ (*), $p < 0.01$ (**); ns: not significant. Source data are provided as a Source Data file.

tumor fibroblasts, macrophages and other non-malignant cells. This approach has been used by others to identify conserved CNVs amongst different prostate tumors that drive tumor growth[46]. Here, we used it to search for CNVs involving genes or groups of genes conserved across the three tumors. To minimize the chances of missing any shared CNVs, we increased the sensitivity of the algorithm so that a 20% false discovery rate was accepted. Under these conditions, several regions were flagged as CNVs in macrophages in all samples (Fig. S6),

which most likely represent false discovery due to high expression of multiple closely-linked genes (e.g. the MHC locus on chromosome 17)[47]. Under such conditions, we did not observe any conserved inferred CNVs in tumor cells from the three albino tumors (Fig. S6). We did observe a putative copy number gain in one tumor in chromosome 6. In addition to carrying the *Braf* gene, chromosome 6 carries several genes that are highly expressed in the LNM and tumor populations (*Aqp1, Col1a2, Cav1, Cav2, Ptn, RaRes2, Fkbp9, Actg2, Ybx3, Mgp, Sox5,*

*Kcna1*, and *Ptms*). Thus, observations in chromosome 6 could be skewed by the high expression of genes characteristic of the LNM and tumor populations, representing an artifact similar to what was observed with macrophages[47]. In summary, these results fail to identify an inferred CNV that affects all three tumors examined.

## LNM cells persist after tumor transplantation

Single cell RNA sequencing identified LNM cells as a population present at low levels in normal skin, enriched in *Braf*[CA/+] skin (Fig. S4E) and highly enriched in tumors (Fig. S4G). To assess whether LNM cells are an integral part of the tumor microenvironment, we subjected tumors to serial transplantation, using an approach developed for propagating patient derived xenografts[48]. *Braf* mutant (albino *Braf*[CA/+] and black *Braf*[CA/+] *Pten*[Δ/+]) tumors were readily transplantable, with the highest rate of success occurring when immune deficient (NSG, Table 1) hosts were used. Interestingly, the time delay to initiate the growth of melanoma tumors shortened after rounds of serial transplantation (Fig. S7A), consistent with enrichment for tumor generating populations. Single cell transcriptomics was used to quantify numbers of cells of different types after different rounds of transplantation (Fig. 3a).

After integrating single cell data from transplants and the primary tumors from which they derived (using Harmony[49], Table 1), and clustering by gene expression, we identified four NC-derived clusters: a principal tumor cluster, a melanocyte cluster, and two LNM clusters (Fig. 3b–d, S7B, Supplementary Data 16). The principal tumor cluster shared many of the same markers as the parental tumors, such as *S100b*, *Postn*, and *Mcam* (Fig. 3c). The "LNM1" population in the transplanted tumors had a similar gene expression profile as the LNM cells in the parental tumors (*Aqp1*, *Igf1*, *Sema3b*) (Fig. 3c). We also identified a separate "LNM2" population, which had decreased expression of some of the markers of parental LNM cells (*Aqp1*, *Ptn*) and expressed some marker genes that were not present in the parental LNM population (*Spp1*, *Igfbp3*). The proportion of LNM cells increased after successive rounds of transplantation (Fig. 3b–e, Fig. S7C), from 10.7% in round 1 to 95.6% in round 2, with the majority of the expansion attributable to LNM2. Of note, LNM2 cells were nearly absent from primary tumors in immunocompetent mice, indicating that their abundance maybe somehow influenced by the immuno-compromised environment. When we calculated the Euclidean distance between all NC-derived cell types we observed that both tumors and melanocytes are slightly closer to LNM2 than LNM1, and LNM1 and LNM2 cells were equidistant from each other (Fig. S7D, Supplementary Data 23). We note that *Aqp1* is expressed in 60% of endothelial cell populations as previously described[50], but also in 95% of LNM1 cells, 34% of LNM2 cells, but only 3% of tumor cells (Fig. 3e, Supplementary Data 21). *Sox2*, on the other hand, is expressed in LNM1, LNM2 cells, and tumor cells, but not in melanocytes or other cell lineages in our dataset. Thus, despite the observed gene expression differences between LNM1 and LNM2, these populations shared the expression of *Sox2* and *Aqp1*, and these markers were not co-expressed in other cell populations in skin.

## A role for *Sox2* in melanoma initiation

We noted that LNM cells were specifically enriched in the expression of the stemness-associated transcription factor *Sox2*, with >30% of LNM cells expressing *Sox2* (Fig. 3c–e, Fig. S7E, Table 1). *Sox2* is also detected in human melanomas[51] and has been described as enriched in melanoma-initiating cells[52], but studies in mouse models thus far failed to support a role for it in melanoma formation. For example, it has been reported by other groups that, in mice bearing either combined *Braf*[CA/+] *Pten*[Δ/Δ 53] or *Nras*[Q61K/+] *Ink4a*[Δ/Δ 54] mutations, loss of *Sox2* did not prevent tumor formation. When we crossed conditional alleles of *Sox2* (*Sox2*[fl]) into the *Braf*[CA/+] *Pten*[Δ/+] background, however, we observed a different result. When *Braf*[CA/+] *Pten*[Δ/+] *Sox2*[fl/fl], *Braf*[CA/+] *Pten*[Δ/+] *Sox2*[fl/+], *Braf*[CA/+] *Pten*[Δ/+] *Sox2*[+/+] mice were treated with tamoxifen at p2-4, a

marked *Sox2* gene dosage-dependent decrease in tumor incidence was observed (Fig. 3f, Fig. S7F). Immunohistochemistry demonstrated that tamoxifen treatment successfully deleted *Sox2* expression in tumor cells (Fig. S7G, Table 1). As LNM cells express the highest levels of *Sox2* and *Sox2* expression is largely absent from melanocytes ( <1%) and present at a low level in tumors (17%), these results are consistent with the contention that the *Sox2* expressing LNM cells may be the cell that initiates *Braf*[CA/+] *Pten*[Δ/+] tumors.

## Comparative analysis of single-cell RNA-seq data between mouse and human tumors

Unlike mouse models, in which oncogenic mutations are usually introduced simultaneously, the order in which mutations (as well as other heritable changes) arise in human melanomas is unknown, and potentially differs from individual to individual. This prompted us to compare the cellular states that we detect in *Braf*[CA/+] and *Braf*[CA/+] *Pten*[Δ/+] mouse melanomas with those described for human melanoma samples, several of which have now been subjected to analysis by single cell RNA sequencing[18], as well as those identified in other mouse melanoma tumors. Although comparison of gene expression states across species can be problematic, we took advantage of the fact that a variety of gene expression signatures have been proposed as markers of human melanoma cell states[55]. To facilitate the mapping of agreement with these signatures onto our scRNAseq data, we developed a "membership score" pipeline that weights the contributions of genes according to their degree of differential expression among the NC-derived cells in our samples (see Methods). Given a set of genes, and a collection of cells, the method assigns each cell a score representing an aggregate of how far above or below the average each gene's expression is. To minimize batch artifacts, the NC-derived cells of each tumor genotype (albino *Braf*[CA/+], albino *Braf*[CA/+] *Pten*[Δ/+], and black *Braf*[CA/+] *Pten*[Δ/+]) were analyzed separately. In Fig. 4a and Fig. S8A–C, cells are displayed on individual UMAPs, and five clusters—principal tumor cells, proliferating tumor cells, melanocytes, Schwann cells, and LNM cells—are highlighted. Membership scores were calculated on a cell-by-cell basis for signatures (Supplementary Data 24-25) that have been reported for treatment-naïve human melanoma biopsies[55] (including patients with both primary and metastatic human melanomas); patient-derived xenografts grown in immune deficient mice and subjected to *Braf*-inhibitor treatment[22]; melanoma cell lines (commercially available and stepwise CRISPR-edited)[19,56]; and other mouse melanoma models (*Nras*[Q61K/+] *Ink4a*[Δ/Δ], *Braf*[CA/+] *Pten*[Δ/Δ 18], and stepwise CRISPR-edited[56]).

Mapping these scores onto the UMAP plots in Fig. 4 revealed some similarities between the cell types identified in our mice and these signatures. Specifically, the principal tumor cells had features resembling the "neural crest-like", "antigen-presenting", and "mesenchymal" signatures of human melanomas; the "invasion" signature of human xenografts; the "mesenchymal" and "partial-EMT" signatures of *Braf*[CA/+] *Pten*[Δ/Δ] mouse tumor cells, the "stem-like" signature of *Nras*[Q61K/+] *Ink4a*[Δ/Δ] mouse tumor cells; the "EMT" signature of CRISPR-edited in-vitro and in-vivo tumor cells; and the "mesenchymal" signature in melanoma cultures (Fig. 4b, Fig. S8E). In contrast, LNM cells most closely resembled the "Interferon-alpha-beta response", "mitochondrial", and "stress" sets of human melanomas; the "intermediate" signature in melanoma cultures; and the "neural crest", "neural crest stem cell like", and "Interferon/TGF" signatures of xenografts and other mouse models (Fig. 4c, Fig. S8F), signatures that overlapped only weakly with principal tumor cells.

Although both LNM and principal tumor cells expressed pigmentation genes at low levels, there was scant pigment visible on the surface of black *Braf*[CA/+] *Pten*[Δ/+] tumors (Fig. 1c), a finding also reported by others[57], which motivated us to examine whether gene expression of any of the cells in these tumors aligned with the *Mitf*-high pigmentation signature that has been identified in other mouse and human

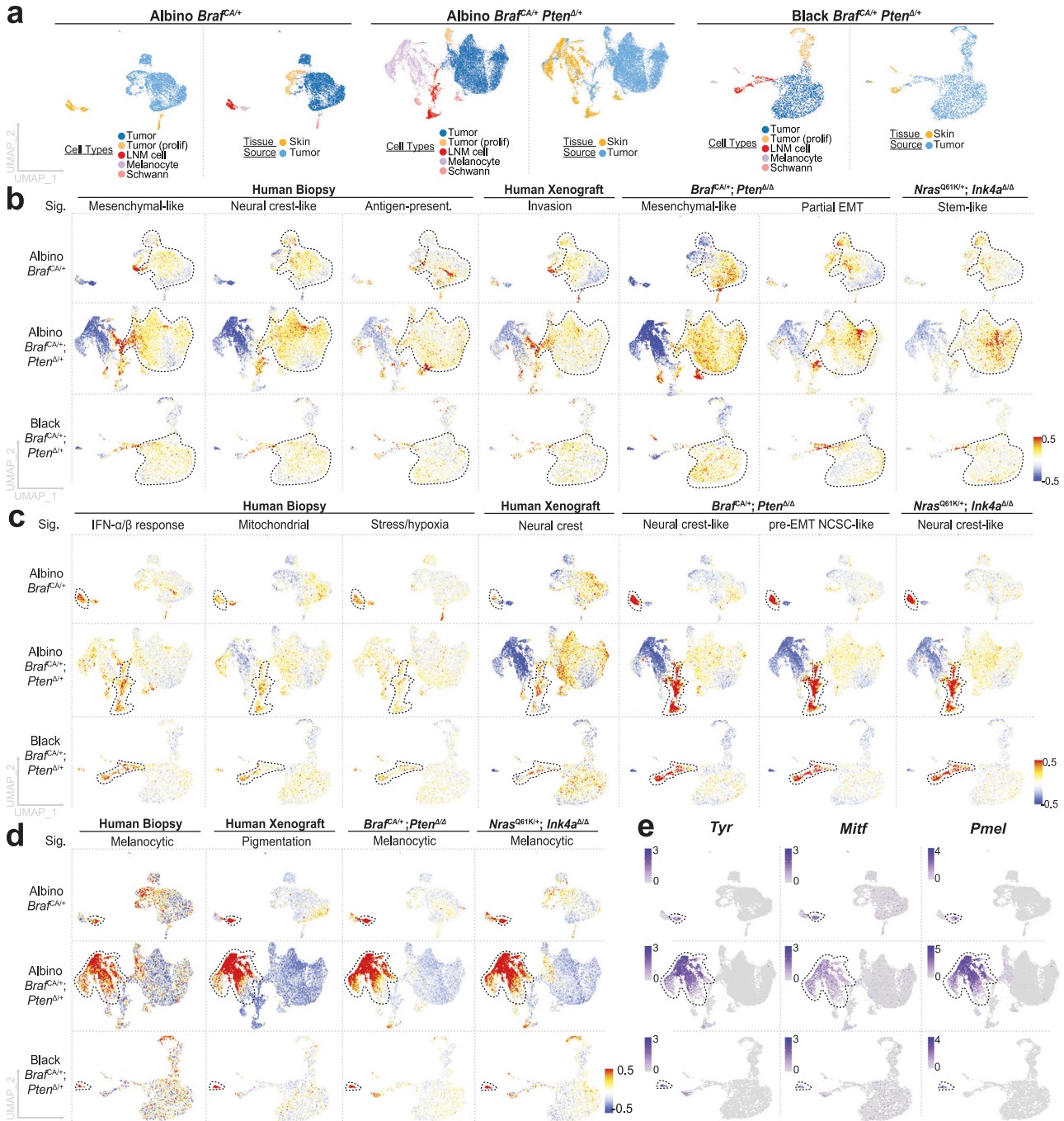

**Fig. 4 | Signatures of NC-derived populations conserved in mouse and human melanoma. a** The tissue source distribution of NC-derived clusters from three genotypes: Albino *Braf*^CA/+ (21,600 cells), Albino *Braf*^CA/+; *Pten*^Δ/+ (5973 cells), and Black *Braf*^CA/+; *Pten*^Δ/+ (3926 cells), is shown. Prolif: proliferating. **b–d** Every cell was assigned a score for how well its gene expression fits a published gene expression signature[18,22,55], relative to the other NC-derived cell types (see Methods and Supplementary Data 24) and these were overlaid on the UMAPs for each of the tumor genotypes (color bar denote membership agreement level; red=strong agreement; blue = poor agreement). Gene signatures that aligned primarily with principal tumor cells are shown in (**b**) those that aligned strongly with LNM cells are shown in (**c**) and those that aligned with melanocytes are shown in (**d**). Black dashes outline cell-types in the individual datasets. **e** Feature plots of expression of selected pigmentation genes in the three tumor datasets. EMT: Epithelial-mesenchymal transitions. NCSC Neural crest stem-cell, Sig signature. Color bar denotes expression level.

melanomas. Membership scoring showed that *Mitf*-high "melanocytic", "melanocyte", "Interferon/p53", "Myc/mTORC1/OxPhos", "OxPhos", and "β-catenin/MITF" signatures identified by others in human biopsies, xenografts, cell cultures, and pigmented mouse melanomas were observed mainly in skin melanocytes, only very weakly detected in principal tumor cells, and essentially absent from LNM cells in our tumors (Fig. 4d, e, Fig. S8G).

## Prediction of cell state dynamics between LNM cells and principal tumor cells

Despite being distinct from principal tumor cells, several characteristics of LNM cells suggest they play a role in tumor formation and growth—for example, the importance of *Sox2*, which is most strongly expressed in LNM cells (Fig. 3c), in tumor formation (Fig. 3e); the persistence of LNM cells during serial

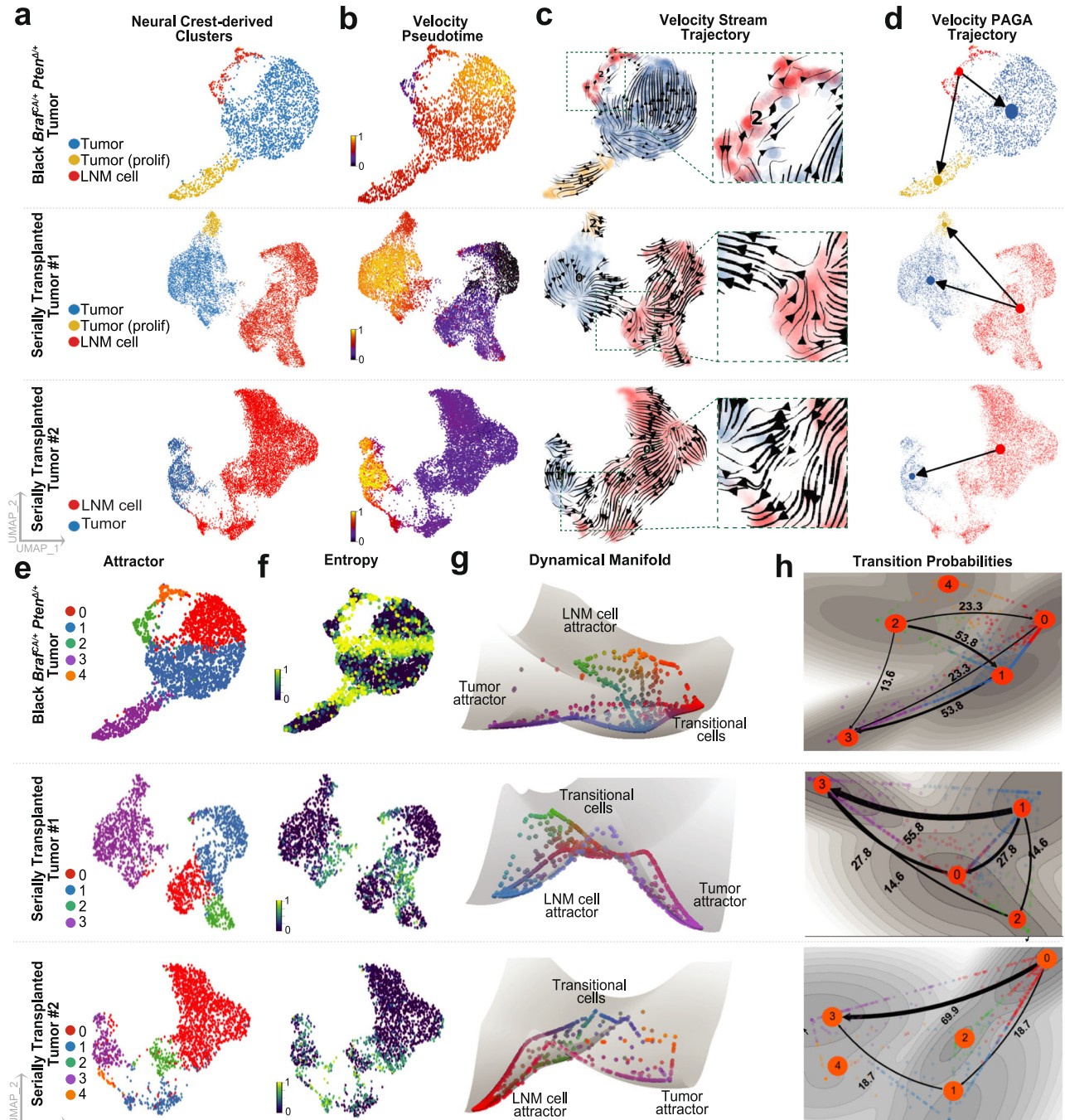

**Fig. 5 | Evidence that LNM cells are transitional cells. a** As described in Fig. 3a, tumors derived from Black *Braf*$^{CA/+}$; *Pten*$^{\Delta/+}$ were serially passaged by transplantation into NSG (immune deficient) mice. Tumors from one parental Black *Braf*$^{CA/+}$; *Pten*$^{\Delta/+}$ genotype mouse and two serially transplanted NSG mice (derived from a single tumor) were subjected to scRNA-seq, analyzed separately, and the neural crest-derived cells were sub-clustered accordingly (parental black *Braf*$^{CA/+}$;*Pten*$^{\Delta/+}$ tumor: 2686 cells; serially transplanted NSG tumor #1 and #2: 10,662 and 10,965 cells respectively). **b–d** RNA velocity analysis by *scVelo* predicts fate decisions of individual cells in A using (**b**) Velocity pseudotime (color bar denotes pseudotime progression), (**c**) Velocity embedding stream trajectories (note the LNM-to-tumor

transition in insets), and (**d**) Partition-based graph abstraction (PAGA), a topology clustering trajectory method with arrows summarizing directionality from LNM to tumor cells. **e** Attractors (stable states) were identified by applying *MuTrans* to the same datasets. **f** Larger entropy values suggest a more transient cell state. Color bar denotes entropy level. **g** The dynamical manifold constructed by *MuTrans*, with potential wells representing stable attractors, and individual cells mapped onto the landscape. **h** Transition path analysis calculates predicted relative rates of transition between stable attractor states. Note the transition path from the LNM cell attractor to Tumor attractor. The numbers along paths indicate the proportion of total transition flux, with larger values suggesting higher likelihood of transition.

transplantation (Fig. 3a–d); and the expression within LNM cells of gene signatures associated with human and mouse melanoma cell states (Fig. 4c, Fig. S8E). Two approaches were taken to test the possibility that LNM and principal tumor cells represent potentially interconverting states.

First, we applied RNA velocity analysis, using *scVelo*[58] (Table 1), to the NC-derived cells of individual tumors (Fig. 5b–d, Supplementary Data 26). RNA velocity uses levels of spliced and un-spliced transcripts to infer cell state dynamics, ordering cells based on the principle that high proportions of unspliced reads indicate recently induced genes

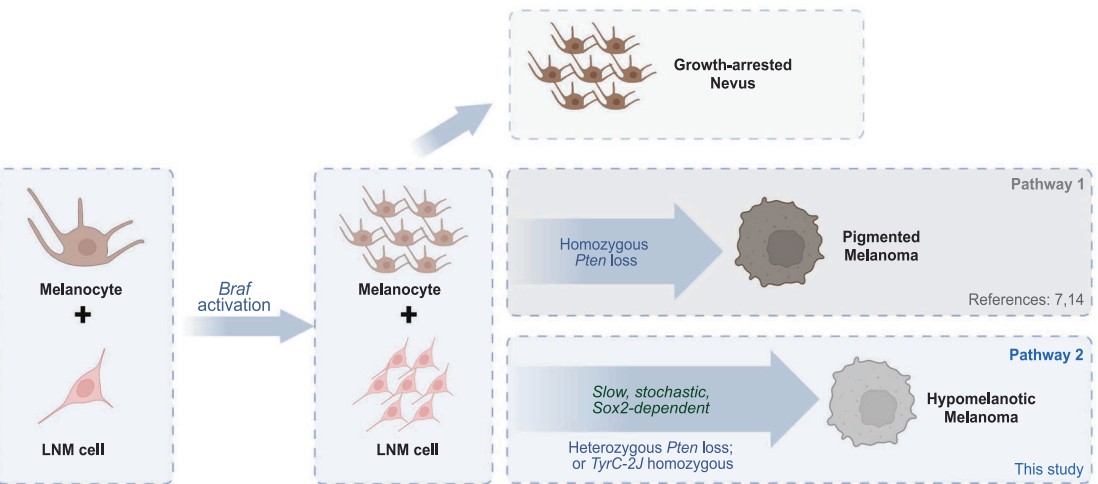

**Fig. 6 | Proposed pathways to melanomagenesis.** Skin contains both pigmented melanocytes and rare, poorly-pigmented LNM cells. Upon *Braf* activation, both populations expand, but melanocytes arrest as nevi, unless loss-of-function mutation in tumor suppressor gene *Pten* allows them to develop rapidly into pigmented melanomas (Pathway 1). Otherwise, *Braf*-activated LNM cells can undergo rare, stochastic transitions to produce scantly pigmented melanomas (Pathway 2). The probability of such transitions depends on host pigmentation status, *Sox2* expression, and whether both alleles of *Pten* have become inactivated. LNM cells persist in both types of tumor. Created in BioRender. Shiu, J. (2025) https://BioRender.com/pf330qi.

while low proportions indicate genes recently turned off. We also analyzed cells using *MuTrans* (Table 1), a pipeline that uses dynamical systems concepts to infer transient cells and cell-fate transitions from snap-shot single cell transcriptome datasets[59] (Fig. 5e–h, Supplementary Data 27). To minimize batch-related artifacts, analyses were performed separately on individual black *Braf*[CA/+] *Pten*[Δ/+] tumor samples. These tumors contained a higher fraction of LNM1 cells but only rare LNM2 cells. To increase the probability of detecting transitions, we also analyzed two sets of tumors obtained after rounds of serial transplantation (Fig. 3b–d). These tumors contain a high fraction of LNM2 cells and rare amounts of LNM1 cells.

In every sample analyzed, both RNA velocity and MuTrans independently supported the conclusion that transitions occur between LNM cells and principal tumor cells (Fig. 5b–d, Fig. 5h). Among all possible transitions identified by MuTrans, the path from cells that express high levels of *Aqp1* to primary tumor was also consistently identified as the most likely of all possible transition paths (Fig. 5b–d, Fig. 5h).

## Discussion

Genetic heterogeneity, epigenetic modifications, and a panoply of cell states are characteristic of melanoma[23], yet the background rate of mutation in human skin[60,61] makes it difficult to ascertain relationships between cell state landscapes, individual mutations and non-genetic factors. In addition, melanomas in individual patients can differ markedly in histological characteristics, and can include the absence of pigmentation[62] as well as the expression of different degrees of "neural" phenotype[63].

Mouse models have thus played an important role in providing insights into how melanomas arise[14]. In mice, combining gain of function oncogene mutations (*Braf* and *Nras*) with homozygous loss of function mutations in tumor suppressor genes (*Cdkn2a, Pten, Ink4a*) provides an efficient route to the rapid generation of melanoma[64,65]. Here we focus on melanomas generated either solely by *Braf* activation, or by *Braf* activation in combination with heterozygous loss of function of *Pten*. Such melanomas arose slowly and rarely, with kinetics indicating the requirement for a low-probability, random event. Interestingly, the incidence data imply that, even though tumor formation in response to *Braf* activation alone is much more probable in albino than black mice (essentially absent in the latter, although

occasionally reported by others[9]), when one allele of *Pten* is also inactivated, tumors arise somewhat sooner in black than albino mice (Fig. 1b). Taken together, the data suggest that the albino allele can exert both positive and negative effects on tumor initiation in the models described here.

The tumors produced by *Braf* activation in combination with heterozygous loss of function of *Pten* (retaining expression of the remaining *Pten* allele) grew slowly and were hypomelanotic. The LNM cells described here overlapped substantially in gene expression with very lowly-pigmented "neural crest" and "neural crest stem cell-like" signatures seen in other mouse and human tumors. Interestingly, in human xenograft models, this slow-growing state is seen in substantial numbers only after treatment with *Braf* inhibitors. It has been proposed that these cells serve as an intermediate on the way to a final, *Braf*-inhibitor resistant tumor cell type[22,66]. Based on comparisons with those and other studies, we propose a model that describes how tumors can arise in melanocytes with *Braf* activation alone or *Braf* activation in combination with heterozygous loss of function of *Pten* (Fig. 6).

The starting point of the model is the observation that normal skin contains both pigment-producing melanocytes and LNM cells—the latter potentially corresponding to various types of neural crest or melanocytic precursor cells that have been described in normal skin[36,67]. These cell types may exhibit a precursor-product relationship, or even interconvert, but this hypothesis remains to be tested. Introduction of an activating *Braf* mutation into melanocytes and/or LNM cells is sufficient to drive the rapid formation of growth-arrested nevi. In mice at least, *Braf* activation also leads to an expansion of LNM cells (Fig. S4E). A rare, stochastic event could then allow for the initiation of slow-growing, hypomelanotic tumors, potentially from *Braf*-transformed LNM cells, whose phenotype such tumors resemble. It could involve a de novo mutation, but evidence for any mutation that was consistent across all tumors, or known to play a role in tumor initiation in melanoma, was not found. Although the molecular nature of that event is as yet unknown, it can be accelerated in the context of a *Tyrosinase*-deficient genotype, or the loss of a single allele of *Pten*, suggesting it is sensitive to even subtle changes in the growth properties of cells. It is possible that the stemness-associated transcription factor Sox2, which is required for tumor formation by this route, itself plays a role in the transition to malignancy, however it is also possible that *Sox2* is simply required for LNM cell growth or survival.

It is important to note that, when human xenografts were subjected to *Braf* inhibition, a distinct subset of cells survived and acted as a precursor to the development of inhibitor-resistant tumors. This subset was described as slow growing, poorly-pigmented, and characterized by a NC-like signature resembling the gene expression pattern of LNM cells (for example, such cells are marked by *Aqp1*). Although they were described as arising in response to inhibitor treatment, it seems likely these cells pre-existed as a minor population; indeed, a minor population with signatures overlapping with LNM cells can be found in primary human melanomas (Fig. 4c), and *Aqp1*+ cells are observed histologically (Fig. S7G, S8D)[55]. Here, using two different approaches (*scVelo*, *MuTrans*) in primary and transplanted tumors, we demonstrate that the LNM state may serve as an initial, or ongoing, source of tumor cells.

As for the question of whether mutation-dependent process is required for tumor initiation in the albino *Braf*^CA/+ tumors, we did our best to characterize any mutations that were shared among tumors. There are scenarios we cannot rule out. For example, we cannot rule out the possibility that multiple different mutations might exist that have similar effects in different tumors, or that certain kinds of mutations were missed, nor could we be certain that any mutation that we observe at high frequency in a tumor did not arise after tumorigenesis and, for reasons having to do either with a fitness effect or neutral drift, came to dominate the tumor cell population. One difficulty in distinguishing these possibilities is that each method to identify mutations has caveats: short-read sequencing misses many structural variants; the power of bulk methods is diminished by the presence of non-tumor cells in tumor samples; inference of CNVs from gene expression data can produce false positive results when groups of nearby genes exhibit cell-type specific expression; and methods designed to identify variants with high confidence (i.e. with few false positives) are not necessarily well-powered statistically to rule out the presence of variants (i.e. display few false negatives).

Despite these caveats, we do report that no established melanoma driver gene, besides *Braf*, is mutated in these tumors, which we believe to be important. Furthermore, we show that tumor incidence in the albino *Braf*^CA/+ model behaves as a single-step stochastic process with a half-time of ~33 weeks (Fig. 1). As we and other have reported previously[9,11,12], *Braf*^CA -transformed melanocytes become growth arrested within about 2 weeks of *Braf*^CA activation in vivo, implying that whatever stochastic processes is leading to tumor formation likely takes place in non-dividing cells. In contrast, while de novo genetic variants are not conserved between the tumors, the cell state and cell state transition proposed to initiate melanoma tumors was seen in every case studied here.

Melanoma can be cured if detected early and removed[68], but it is often difficult to distinguish from benign lesions using clinical examination, histology[69], or DNA alterations[4,70]. It is intriguing to note that, in the present study, LNM cells accumulated in *Braf*^CA/+ *Pten*^fl/+ and *Braf*^CA/+ mice even in skin in which tumors had not yet developed (or could not do so). The detection of these cells in biopsies may represent a clinically useful marker of malignant potential as it appears that they are conserved across multiple different melanoma models in mice and humans. Furthermore, to the extent that these cells may ultimately act as a progenitor to tumor cells, they may also represent a target for development of preventive therapies. If the malignant transformation of these cells involves an epigenetic, rather than a genetic change, a wider range of therapeutic modalities could be explored than are usually considered in skin cancer prevention, which is currently focused on reducing mutation burden.

## Method

All of the animal studies presented here were performed in compliance with institutional guidelines and approved by Institutional Animal Care and Use Committee (IACUC) at University of California, Irvine prior to experiment initiation (protocols: AUP17-230, AUP-20-161, AUP-23-116). Similar numbers of male and female mice were used in all the described experiments, and the sex of the mice used has been included in the data source file. For mice with tumors, animals were euthanized when tumors exceeded 10% of body weight or when mice exhibited loss of body condition. Tumor size was monitored with calipers, and mice were euthanized before tumors reached a size of 2000 mm³. No statistical method was used to predetermine sample size. No data were excluded from the analyses, and sample size was determined based on prior experience and sample availability. No patient samples were used to generate sequencing data or perform other analyses. Only publicly available human single cell sequencing data was used for this work[55].

### Generation of *Braf*^CA/+ melanoma and nevi bearing mice

Conditional allele *Braf*^CA, *Tyr::CreER*, and/or *Pten*^lox4-5/+ mice (RRID:MGI:5902125, from Dr. Martin McMahon) were backcrossed with C57BL/6 J (JAX 000664) for greater than five generations. Conditional allele *Braf*^CA, *Tyr::CreER*, and/or *Pten*^lox4-5/+ congenic C57BL/6 J mice were then crossed with Albino B6(Cg)-*Tyr*^c-2J/J (JAX 000058) to create Albino *Braf*^CA/+ melanoma and nevi bearing mice. Mice were genotyped by PCR as previously described (Bosenberg et al., 2006; Dankort et al., 2007). The primers used in this study are: Braf forward 5′-TGAGTATTTTTGTGGCAACTGC −3′, Braf reverse 5′-CTCTGCTGG GAAAGCGCC −3′, Pten forward 5′-AAAAGTTCCCCTGCTGATGATTTGT −3′, Pten reverse 5′-TGTTTTTGACCAATTAAAGTAGGCTGTG −3′. Cre WT_forward 5′- TTC CCA CAC TTA ACA GCC CCA −3′ and Cre WT_reverse 5′- GGA CGT GTG GAG GGA TCG T −3′. Cre Tg_forword 5′- TTC CCA CAC TTA ACA GCC CCA −3′ and Cre Tg_reverse 5′- CCC ACA TCA GGC ACA TGA GT −3′ (Table 1). Albino mice were identified by their coat color. All mice used in the study were heterozygous for the *Tyr::CreER* allele, which was verified by primers that could distinguish the wild type and mutant alleles[71]. The breeding schemes used to generate Albino *Braf*^CA/+ and *Braf*^CA/+*Pten*^Δ/+ mice are shown in Fig. S1C; based on these we estimate the chances that tumor development in albino *Braf*^CA/+ mice could have depended on the presence of an unknown modifier gene present in founder albino mice to be small. Specifically, if an unlinked homozygous modifier had been present in both founder animals, we calculate the probability of it appearing in homozygous form in all tumor bearing mice at < 1/10,000, and the probability of it appearing in heterozygous form at <7.5%.

Topical 4-hydroxytamoxifen (4-OHT; 25 mg/mL or 75 mg/mL in DMSO; 98% Z-isomer, Sigma-Aldrich, Table 1) was administered to pups on their back at ages P2-P4. Mice were euthanized if the volume of their tumors exceeded 10% of total body volume, if tumors were significantly ulcerated, if mice were moribund, if they lost weight, if they were lethargic, or if they were unable to ambulate. All mouse procedures were approved by IACUC at University of California, Irvine.

### Tissue harvest for whole-genome sequencing

Melanoma-bearing albino *Braf*^CA/+ mice (*n* = 3, aged postnatal day 90 to 325) were euthanized, shaved, and depilated. Tumor (25 mg), skin (25 mg), and spleen (10 mg) were then collected from each mouse and cut into small pieces with a scalpel, and the fat scraped off from the underside of the skin. Collected tissues were then stored in −80°C freezer until sample preparation day. DNA materials were extracted using the Qiagen blood and tissue kit, following protocols described for purification of total DNA from animal tissues. Briefly, approximately 25 mg of tissue was obtained and digested with ATL buffer and proteinase K. Ethanol was added to the lysate and passed through a DNeasy spin column. DNA was eluted with AE buffer.

### Library preparation for whole-genome sequencing

Library construction was performed using the NEXTflex Rapid DNA-Seq kit v2 and the NEXTflex Illumina DNA barcodes. Using the Covaris S220, 50 ng of gDNA was sheared using settings to target 400 bp. The

sheared gDNA was end repaired and adenylated. The reaction mixture was cleaned up using AMPure XP magnetic beads and Illumina barcoded adapters were ligated onto the blunt-end/adenylated product. The adapter ligated product was cleaned using AMPure XP beads and then amplified for adapter ligated products using 5 cycles of PCR. The resulting library was cleaned with AMPure XP beads with double sided size selection and then quantified by qPCR with Kapa Sybr Fast universal for Illumina Genome Analyzer kit. The library size was determined by analysis using the Bioanalyzer 2100 DNA HighSensitivity Chip. The library was sequenced on the NovaSeq 6000 Sequencer, using an S4 flowcell chemistry and PE100 cycles with additional cycles for the index read. Sequences were obtained as paired-end 100 bp reads. The NovaSeq control software was v1.6.0 and the real time analysis software (RTA v3.4.4) converted the images into intensities and base calls. Post-processing of the run to generate the FASTQ files was performed at the UCI Institute for Genomics and Bioinformatics (UCI IGB).

### Estimation of *Braf* allele recombination frequency

In the mice used in this study, expression of constitutively active *Braf* was achieved through the recombination of an engineered allele in response to a melanocyte lineage-specific *Cre*[24]. In cells that do not undergo recombination, two loxP sites are present, each flanked by unique sequences (unrecombined loxP), whereas in cells that have recombined, a single loxP site remains, bringing together the flanking sequences from each of the two original sites (recombined loxP). In principle, one can use genome sequence data to estimate the fraction of cells in a tumor sample that underwent *Braf* recombination by examining the proportions of recombined and unrecombined loxP sequences in any sample.

Briefly, for each tumor, all sequences were extracted that matched the entire 34 bp loxP sequence and also contained at least two nucleotides at either end, allowing unambiguous assignment of recombined vs. unrecombined status. The fraction of recombined cells in each tumor sample was then calculated as the number of recombined sequences divided by the sum of the number of recombined sequences and half the number of unrecombined sequences (half because unrecombined cells have twice the number of loxP sites as recombined ones). In the three tumors analyzed here, the total number of loxP sequences used in the calculation was between 106 and 194. As a control, we also assessed recombination status in spleen and skin. In spleen, no recombined loxP sequences were detected in any tumor, as expected. In skin, no recombined sequences were detected in two cases, while two were detected in a single sample, implying a recombination frequency of 1.3%; this is consistent with the fact that melanocytes (which had the potential to recombine) are present at about this frequency in normal skin.

### Whole genome sequencing Single Nucleotide Variant (SNV) analysis

For SNV analysis, raw reads were transferred, and quality analyzed using *FastQC* tool (v0.11.9). Low quality bases and adapter sequences were trimmed using *Trimmomatic* (v0.39)[72](Table 1). Trimmed reads were then aligned to the mouse reference genome (build mm10) using the Burrows-Wheeler Aligner (*BWA mem*, v0.7.12)[73]. Duplicated reads were removed using *Picard tools MarkDuplicates* (v1.130). Local realignment and base quality recalibration was done on each chromosome using the Genome Analysis Toolkit (*GATK* v4.0.4.0) best practices. SNVs and indels were detected and filtered using GATK with HaplotypeCaller function. The output files were generated in the universal variant call format (VCF).

Somatic mutations were identified and filtered using *Mutect2* (v4.1.3.0)[74] and *bcftools* (v1.15.1)[75]. To estimate the fraction of DNA in each sample that was derived from *Braf*^CA/+^-expressing cells (as opposed to fibroblasts, immune cells, etc.), we identified all reads that

mapped to the loxP sequence and that included at least three bases on either side. From the sequences up- and downstream of loxP we were able to uniquely assign each read to one of three categories: an unrecombined 5' loxP site, and unrecombined 3' loxP site, and a recombined loxP site. Unrecombined cells should contain one each of the sequences of the first two types, whereas recombined cell should contain a single sequence of the third type. Thus, the ratio of the observed number of sequences of the third type to the average of the number of sequences of the first two types provided an estimate of the ratio of recombined to unrecombined cells.

### Variant annotation with annovar

Mutect2 called VCF files from each sample were first filtered with bcftools (v1.10.2) to retain variants with "PASS" flag. The filtered VCF files were then indexed, normalized with bcftools, and then annotated using *Annovar* (version date 2020-06-07, Table 1) gene annotation function (*annotate_variation.pl*). This identified variants as compared to the RefSeq gene database that were a consequence of stop loss or stop gain or that hosted non-synonymous mutations or variable splice sites. The resulting variants were annotated as far as their type, i.e. exonic, UTR5/UTR3, intronic, or intergenic.

### Structural variant calling Copy Number Variation (CNV) analysis

For structural variant calling, *BreakDancer* (v1.1.2, Table 1)[76] was used to analyze mapping results from whole genome sequencing and classify structural variations in five categories: deletion, insertion, inversion, intrachromosomal translocation and interchromosomal translocation. Genomic regions with statistically significant amounts of anomalous read pairs (split reads and their depth) are considered supporting structural variations breakpoints. A Poisson model was implemented to incorporate the number of supporting anomalous reads, the size of the region, and the genomic coverage in order to compute a confidence score. The default set of parameters was used for BreakDancer.

We followed the recommended BreakDancer protocols[32] and analyzed the matched tumor, skin and spleen samples for each of the 3 animals. BreakDancer was run 3 times, each with 3 genomes from the same animal. In order to get somatic CNVs, the identified breakpoints were then filtered so that only those from the tumor are preserved (Tumor-skin-spleen). The 3 identified CNV lists from the 3 animals were then annotated with gene symbols if overlapping with a gene body. The 3 resulting gene lists were intersected to identify plausible recurrent somatic CNVs.

### Coverage based Copy Number Variation (CNV) analysis

*CNVkit*[33] (v0.9.11, Table 1), a read depth-based approach, was used to call and visualize copy number variants in tumor samples using the matched normal samples as background. Specifically, CNVkit attempts to detect structural variants based on differences in sequence coverage when compared with a reference sample. Aligned BAM files from both tumor and normal samples were imported to calculate copy number ratios that deviate from the expected by an amount consistent with structural alteration in a fraction of the genomes equal to the (known) tumor cell fraction. Segments assigned copy number (cn) values less than or greater than 2 are potential sites of deletion or duplication. Because CNVkit is prone to making false positive calls, particularly with whole genome sequencing and especially for indels <1 Mb, we also used CNVkit to compare each spleen against the other two spleens (where we do not expect to see high-frequency structural variants). We found that use of a cutoff score (weight) of 100 kept the number of autosomal genes with CNV calls in spleen samples below 101. Under these conditions, tumors 1, 2 and 3 displayed 13.6, 6.9 and 3.5 times as many genes in potential CNV regions, respectively, as their matched spleen samples, but even so the overlap between the three tumors was small and did not involve any genes known to be mutated, deleted or amplified in human melanoma[1] (Supplementary Data 11, 14,

17). Only four genes, *Hdac1*, *Khdrbs1*, *Zcwpw1*, and *Mta3*, overlapped with any CNVs identified by BreakDancer in any tumor, and in each case the overlap was in only one tumor. The breakpoints identified by BreakDancer in three of these cases were supported by only two or three reads, suggesting a low frequency event, and in all cases the intervals predicted by BreakDancer (Supplementary Data 9–10, 12–13, 15–16) differed greatly in size from those predicted by CNVkit (Supplementary Data 11, 14, 17).

## Mutational signature detection with SigProfiler

The COSMIC *SigProfiler*[31] (Table 1) tool was used to explore mutational signatures in each tumor. Filtered VCF files from Mutect2 somatic mutation calling were imported and Single base substitution (SBS) matrices were generated and visualized using the *SigProfilerMatrixGenerator* function in Python (v3.8). Default parameters were used to generate the data reported in Fig. S2B.

## Cell isolation for single-cell RNA sequencing

Nevi-bearing *Braf*^WT^, *Braf*^CA/+^, and *Braf*^CA/+^ *Pten*^Δ/+^ mice were harvested at P30 and P50. Melanoma bearing *Braf*^CA/+^ (albino coat color), *Braf*^CA/+^ *Pten*^Δ/+^ (albino and black coat color), and transplanted tumors were harvested when tumors reached a size that affected the health of the animal, as described above, complying with IACUC regulations. The mice were euthanized, shaved, and depilated. For non-tumor bearing samples, a 2 by 3 cm section of the back skin was retrieved, and the fat scraped off from the underside. For melanoma-bearing samples, the skin sample on top of the tumor piece was also collected, with the mouse-matched adjacent skin next to the tumor (termed "tumor-adjacent skin") as sample-matched control.

The tumor or skin sample was then physically minced, and suspended in a gentle-macs C-tube dissociation buffer (5 mL of RPMI, 50 μL of liberase 0.25 mg/mL, 116 μL of Hepes 23.2 mM, 116 μL of Sodium Pyruvate 2.32 mM, 500 μL of Collagenase:Dispase 1 mg/mL) for 50 min at 37 °C incubation with gentle agitation at a speed of 85–90 rpm. After initial 50 min digestion, 23 μL of DNase I was added for another 10 min of 37 °C agitated incubation and then inactivated with 400 μL of fetal bovine serum (FBS) and 10 μL of EDTA (0.5 mM). The tissue suspension was further dissociated mechanically with GentleMACS by running the setting "*m_imptumor_04.01*" twice. The digested suspensions were filtered twice through a 70 mm strainer and dead cells removed by centrifugation at 300 x g for 15 min. The live cells were washed with 0.04% UltraPure BSA:PBS buffer, gently resuspended in the same buffer, and counted using trypan blue and cell counter.

## Library preparation for single-cell RNA sequencing

Libraries were prepared using the Chromium Single Cell 3' v2 protocol (10X Genomics). Briefly, individual cells and gel beads were encapsulated in oil droplets where cells were lysed and mRNA was reverse transcribed to 10X barcoded cDNA. Adapters were ligated to the cDNA followed by the addition of the sample index. Prepared libraries were sequenced using paired end 100 cycles chemistry for the Illumina HiSeq 4000. FASTQ files were generated from Illumina's binary base call raw output with Cell Ranger's (v2.1.0; RRID:SCR_017344, Table 1) '*cellranger mkfastq*' command and the count matrix for each sample was produced with '*cellranger count*'.

## Single-cell RNA sequencing analysis of *Braf*^CA/+^; *Braf*^CA/+^ *Pten*^Δ/+^ mice

47 samples from the genotypes: wild type; *Braf*^CA/+^; *Braf*^CA/+^ *Pten*^Δ/+^; *and Pten*^Δ/+^ were aggregated using the Seurat built-in "*Merge*" function to produce one count matrix. Downstream bioinformatic analysis was conducted using Seurat[77] (Table 1). For quality control, cells with fewer than 200 detected genes and genes detected in less than 3 cells were discarded. We calculated the percent mitochondrial gene expression

and kept cells with less than 15% mitochondrial gene expression, and cells with fewer than 4000 genes per cell. A total of 345,427 cells from the whole skin samples passed the quality control. Each cell was then normalized using *scTransform*. In the final preprocessing step, we regressed out cell-cell variation driven by mitochondrial gene expression and cells with few genes captured using the number of detected UMI. To identify cell-type clusters, principal component analysis using highly variable genes, Louvain clustering, and visualization with Uniform Manifold Approximation and Projection (UMAP) was used. For generating differentially expressed gene heatmaps, cells were downsampled to 200 cells. The Euclidean distances between clusters was calculated using the average principal component (PC) embedding for each cluster, from the top 10 PCs.

## Copy number variant inference analysis

*InferCNV* package (version 1.20.0, Table 1) was used to infer CNVs from single-cell RNA sequencing profiles from three Albino *Braf*^CA/+^ mice with tumors were used, along with their mouse-matched skin samples. Fibroblasts were used as the reference cell group, whereas melanocytes, LNM cells, tumor cells, endothelial cells, erythrocytes, T cells, and macrophages are used as observation cell groups. To minimize noise from sparse matrices, cells with <6300 UMI counts were filtered out, and the *Braf* gene was removed from the count matrix as the expression of this gene would be altered by recombination. As advised (https://github.com/broadinstitute/inferCNV), cut-off was set as 0.1. All other default parameters were used except for *window_width* = 151 to facilitate visualization and *BayesMaxPNormal* = 0.2 to include only those inferred CNVs with high confidence.

## Immunofluorescence

Formalin fixed paraffin embedded sections were sectioned 6 μm and placed on poly-L-Lysine glass slides (New Erie Scientific LLC, NH). Slides were dried overnight, and paraffin was removed with xylene followed by rehydration. Antigen retrieval was carried out by heating slides to 80 °C for 50 min in 0.1 M Tris-based (pH 9.0) (Vector Lab, CA). Melanin bleaching was performed using 3% H2O2 overnight at room temperature, followed by stop reaction using 1% of acetic acid. True black lipofuscin autofluorescence quencher was then applied for 30 s. Protein block was carried out using either 10% normal goat serum in 1x PBS for 1 hr. Samples were then incubated with primary antibody Pten (1:100) (Cell Signaling Tech, MA, Table 1) overnight at 4 °C, followed by incubations with Alex fluor 594 secondary antibody (1:1000) (Life technology, CA) and 0.5 ng/ml of DAPI for 1 hour at room temperature. Slides were washed with 1X PBS three times before mounting with prolong gold antifade reagent. Slides were viewed using the Keyence BZ-X810 Wide-Field Microscope in the Stem Cell Research Center at the University of California, Irvine. Images were captured at high resolution with the same exposure time.

## Histopathology and Immunohistochemistry staining

For immunohistochemistry, formalin fixed paraffin embedded sections were sectioned around 5 μm–8 μm thick onto poly-L-Lysine coated slides in a tissue water bath at a temperature of 31 to 37 °C. Slides were deparaffinized with Xylene and dehydrated in a series of ethanol washes with increasing concentration. Antigen retrieval was performed with 10 mM citric acid buffer at pH 6.0 for 50 min in a 80 °C water bath, then left to cool overnight at room temperature. All samples were next incubated with the primary antibody overnight at 4 °C. Samples were washed and incubated with the appropriate secondary antibody. Hematoxylin and eosin (H&E) staining was performed using standard histopathological methods[78].

## RNAscope fluorescence in situ hybridization

RNAscope Fluorescence In situ hybridization[79] was performed using the RNAscope Multiplex Fluorescent Reagent Kit v2 (catalog 320293,

Table 1). Briefly, formalin fixed paraffin embedded sections were sectioned around 7 μm thick onto poly-L-Lysine coated slides in a tissue water bath at a temperature of 37 °C. Slides were dried overnight in a 37 °C incubator, baked at 60 °C for 2 h, and then deparaffinized two times with xylene for 5 min, and two times with 100% EtOH for 2 min. After H202 treatment at room temperature for 10 min, samples were boiled for target retrieval for 15 min, and permeabilized using protease at 40 °C for 30 min. Samples were then incubated with Aqp1 (C2) and Sox2 (C3) probes at 40 °C for 2 h. The C2 and C3 signals were detected using Opal-620 (Akoya Biosciences) and Opal-650 (Akoya Biosciences) respectively. Dapi was used to stain the nuclei and prolong gold solution was used to mount the slides. Images were acquired using a Leica SP8 Microscope (25× water objective).

### Congenic tumor transplantation

Melanoma-bearing Albino $Braf^{CA/+}$ and Black $Braf^{CA/+}$ $Pten^{Δ/+}$ donor mice were euthanized, shaved, and depilated when tumors reached around 10% of body weight. After each tumor was collected, it was cut into pieces (weighing around 0.02g–0.06 g) in a cake-slice pattern to preserve tumor structural heterogeneity; this was performed in a biosafety cabinet to ensure a sterile environment. The pieces were then washed twice with 70% EtOH, and once with 1x PBS, each for 10 s. They were then transferred to RPMI + 10% FBS media on ice until transplantation.

NSG recipient mice (JAX 005557, Table 1) were anesthetized using Isoflurane. The tumor piece was transplanted to the back flank skin by cutting a small incision and slipping the tumor tissue in together with 100 μL of Matrigel into the skin. This was done after the mouse was shaved and sanitized with 100% EtOH and chlorhexidine diacetate disinfectant solution. Transplantation shams were also included ($n$ = 1-2 per group) by only transplanting 100 μL of Matrigel to the mouse back. 10 μL of lidocaine was applied to the wound for three consecutive days after the procedure. Tumor size was measure weekly.

### Single-cell RNA sequencing data integration of transplanted NSG tumors

A total of 15 scRNA-seq samples from primary Albino $Braf^{CA/+}$ $Pten^{Δ/+}$ tumors ($n$ = 4), tumor transplantation series 1 (donor: Albino $Braf^{CA/+}$ mice, $n$ = 3; host: NSG mice; 1 round of transplants; $n$ = 2) and Tumor transplantation series 2 (donor: Black $Braf^{CA/+}$ $Pten^{Δ/+}$; host: NSG; 2 rounds of transplants; $n$ = 2 for each) were merged using Seurat Merge function and integrated with Harmony[49] (v1.2.0, Table 1).The remaining pre-processing and bioinformatic analysis were performed as described above. The Euclidean distances between clusters was calculated using the average Harmony embedding for each cluster, from the top 30 Harmony space embeddings.

### Generation of $Braf^{CA/+}$ $Pten^{Δ/+}$ $Sox2$-deficient mice

$Sox2$-floxed mice were purchased from Jackson laboratories (STOCK Sox2tm1.1Lan/J strain#:013093, Table 1) and crossed to $Braf^{CA-fl/+}$ $Pten^{fl/+}$ mice. These mice were backcrossed into a pure C57BL/6 background for greater than six generations to generate $Braf^{CA-fl/+}$ $Pten^{fl/+}$ $Sox2^{fl/fl}$ mice. Induction of tumors in these mice was performed as described above.

### Membership score analysis

To quantify how well cell types identified by single cell RNA sequencing fit a proposed gene expression signature, we first converted gene expression values to modified, corrected Pearson residuals, as described[80]. We then empirically determined estimates for the coefficient of variation of biological gene expression noise of 0.55, 0.65, 0.65 respectively, for Albino $Braf^{CA/+}$, Albino $Braf^{CA/+}$ $Pten^{Δ/+}$, and Black $Braf^{CA/+}$ $Pten^{Δ/+}$ samples, respectively. Residuals were then standardized, essentially converting them to z-scores. The average of the z-scores for all of the genes in any proposed gene signature was then

calculated for each cell, and this value was plotted as a color-score on the UMAP diagram of the cells. The compared signatures are in Supplementary Data 24–25. For the human biopsy scRNA-seq dataset comparison[55], the publicly available processed human malignant Seurat object was further subsetted into treatment-naive cells based on "Timepoint" being set to "BT" (Before Treatment), and differentially expressed gene signatures were obtained by running $FindAllMarkers$ in the Seurat pipeline.

### RNA velocity analysis

The $scVelo$ package[58] (version 0.2.5, Table 1) was used for RNA velocity analysis. To estimate the velocity based on unspliced and spliced counts, we applied "$dynamical$" mode which uses differential equations to model the transient dynamics of gene expression and splicing, with other parameters set as default. Pseudotime was calculated by the "$velocity\_pseudotime$" function with default parameters. The streamlines of velocity were visualized using "$velocity\_embedding\_stream$" function with parameter density set as 1. Cell lineage analysis was then performed by Partition-based graph abstraction (PAGA) (provided in scVelo package) based on the calculated velocities.

### MuTrans analysis

For $MuTrans$ analysis (Table 1)[59], we used all cells in the Black $Braf^{CA/+}$ $Pten^{Δ/+}$ sample and subsampled 20% cells from the two transplanted NSG mouse tumor samples. The number of attractors were determined by EPI (eigen-peak index) of MuTrans. The cellular random walk was constructed using cosine similarity of gene expression as the input to MuTrans. The dynamical manifold was constructed using default parameters based on UMAP dimension reduction coordinates as the initial values. Transition gene analysis was performed using the "GeneAnalysis" module in MuTrans, with parameter "$thresh\_de\_pvalues$" set as $5e^{-4}$ and all other parameters kept as default.

### Statistical analysis

Kaplan Meier survival curves were generated, and significance assessed using the log-rank test, an unpaired $t$-test, or a two-way ANOVA test. Other data were analyzed by GraphPad Prism6 statistical analysis software using an unpaired $t$-test or two-way ANOVA test. Significance levels were as indicated. For WGS CNV analysis using CNVkit, $p$-values were calculated using two-tailed $t$-test. For scRNA-seq differentially expressed genes, $p$-values were assessed using the default Wilcoxon Rank Sum test.

### Reporting summary

Further information on research design is available in the Nature Portfolio Reporting Summary linked to this article.

## Data availability

The scRNA-sequencing and WGS sequencing data generated in this study have been deposited in the SRA repository under BioProject accession code PRJNA1172319 and in the GEO repository under accession code GSE279468. Processed data of all human scRNA-seq from previously published study[55] is publicly available at KU Leuven RDR [https://doi.org/10.48804/GSAXBN], among which the "Malignant_cells.rds" object was used. Source data are provided with this paper.

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

## Acknowledgements

This work was supported by NIH CA217378 (AKG, ADL), CA288662 (AKG, ADL), CA244571 (AKG), and P30AR075047 (AKG, ADL). HX was supported by CIRM under Award EDUC4-12822. RRV was supported by the UC Presidents fellowship and a FORD Foundation Fellowship. QN was partly supported by supported by NSF grants DMS1763272 and a Simons Foundation grant (594598). This work was made possible, in part, through access to the Genomics High Throughput Facility Shared Resource of the Cancer Center Support Grant (CA-62203) at the University of California, Irvine and NIH shared instrumentation grants 1S10RR025496-01, 1S10OD010794-01, and 1S10OD021718-01. Fluorescent images were generated through the Chao Family Comprehensive Cancer Center Optical Biology Core Facility, supported by P30CA062203 and S10OD028698. Biorender was used for schematics.

## Author contributions

A.D.L. and A.K.G. conceptualized the study. H.X., J.S., C.C., S.S.T. and R.R.V. curated the data. H.X., J.S., C.C., J.W,. P.Z., S.S.T., R.R.V., R.A.E. and A.D.L. conducted the formal analysis. H.X., J.S. and C.C. carried out the investigation. H.X., A.D.L. and A.K.G. wrote the original draft. H.X., J.W., P.Z., A.D.L. and A.K.G. reviewed and edited the manuscript. A.D.L. and A.K.G. acquired funding for the project. A.D.L., P.Z., J.W. and Q.N. developed the methodology. A.D.L. and A.K.G. administered the project.

## Competing interests

The authors declare no competing interests.
