## [Peer Review File · Nature Communications]

Uncovering Minimal Pathways in Melanoma Initiation

Corresponding Author: Dr Anand Ganesan

Version 0:

Reviewer comments:

Reviewer #1

(Remarks to the Author)

Understanding the genetic events that drive melanoma progression has proven challenging due to the high mutational burden observed even in normal skin. Additionally, the presence of BRAF and NRAS mutant melanocytes in benign human nevi shows that MAPK activation is necessary but insufficient to stimulate melanoma progression. Here, Xiao et al. use single cell sequencing to analyze tumor progression in genetically engineered mouse models of spontaneous Braf-mutant melanoma. They identify a distinct cell population characterized by low-pigment, neural, and extracellular matrix (LMN) gene expression. These LMN cells exist in normal skin, expand upon BRAF activation, and persist through multiple passages in mice. Notably, the transcriptional profile of LMN cells closely resembles neural crest-like populations seen during the dynamic, phenotype switching of human and murine melanoma cells. These findings suggest that phenotype switching plays a pivotal role not only in melanoma progression but also in its initiation.

Beyond its implications for melanoma development, this study underscores the significance of non-mutational, epigenomic events in melanoma initiation. Further, the comprehensive single cell sequencing dataset, comprising over 45 samples from one of the most used pre-clinical GEMMs of melanoma, will undoubtedly serve as a valuable resource for the research community.

Several limitations must be addressed to strengthen the scientific rigor, conclusions, and significance of this work. These limitations, include efforts to: 1) determine how the albino allele promotes tumor initiation in specific genetic contexts, and clarify its relevance to fair-skinned individuals who produce pheomelanin—a factor associated with carcinogenesis, 2) clearly define and consistently analyze the LMN1 and LMN2 cell populations as separate entities, 3) refrain from, conclusions based on comparisons with genetically distinct mouse models conducted by other groups, 4) improve transparency, rigor, and accuracy by further refining the analytical methods, data reporting, and the final model.

Specific Concerns

1) Tumor promotion by the albino allele: The authors suggest that albinism contributes to melanoma progression in mice. They suggest that this is similar to evidence that skin color affects human melanoma risk independently of mutations. However, even individuals with light skin tones still produce melanin. Furthermore, in their Pten heterozygous model, albino mice exhibited slower tumor development (Figure 1B). How the albino allele impacts tumor progression in a genotype-dependent manner is never addressed.

2) Analysis of the LMN1 and LMN2: The authors' conclusion that LMN cells increase through successive tumor transplantations raises several questions. Notably, LMN1 cells appear to decrease in frequency with subsequent transplantation whereas, LMN2 cells increase. How the LMN2 population was determined to be LMN1-like is unclear – especially since LMN2 cells lack several key markers defined in the original analysis (e.g., Aqp1, Igf1, Sema3b; Figure 3C). It is difficult to read the Euclidean distance in Supplemental Figure 4C, but it appears that the LMN1 and LMN2 cells are almost equally different from one another as they are from the tumor or melanocyte populations. In the tumor cell projections (Figure 5), sometimes the LMN populations are divided into LMN1 and LMN2 and sometimes they are not. This seems like an important designation given that LMN1 frequently appears to decrease whereas LMN2 numbers increase with tumor transplantation.

Finally, Sox2 appears to be most strongly expressed in LMN1 cells which are lost during multiple rounds of tumor grafting (Fig. 3C). This brings into question the relevance of Sox2 in LMN-dependent tumor progression.

3) Internal animal controls: Due to potential confounding factors such as genetic drift, methodological variations, and animal husbandry practices, comparisons between mouse models should be conducted under identical conditions—especially given their significance to the authors' model. Examples of specific concerns related to this technical limitation are:

In Figure 1C (lower right), only one image documents the pigmented state of *BrafCA/+; PtenΔ/Δ* mice. It appears to show intact hair follicles. Did a pathologist confirm that this was a bona fide tumor and not a melanocytic hyperproliferation, which is known to occur in the *BrafCA/+; PtenΔ/Δ* model?

The authors propose that *Sox2* deletion limits tumor growth in *BrafCA/+; PtenΔ/+* mice but not in *BrafCA/+; PtenΔ/Δ* animals. However, data for the *Pten* null phenotype is not shown to validate replicability of published studies in their lab. Additionally, tumor number may not reflect onset, but differences in aggressive behavior of the melanomas that impact euthanasia timing. Were there variations in the Kaplan-Meier curve or tumor growth rates of *Sox2* null and wildtype mice?

4) Data transparency, rigor, and accuracy

Variant calling: The number of variants detected in this study exceeds what has been reported for similar spontaneous melanoma models in the literature. Notably, many of these changes are observed in genes with fewer than five reads, as indicated in Supplemental Tables 1-6. To address this, it's essential to understand how the minimum threshold for mutation and copy number variant (CNV) calling was determined. Additionally, please provide a key for the headers in Supplementary Table 1, along with clarification on how each normal sample was used during the variant calling process.

Regarding the CNV analysis, it's intriguing that gene amplifications were not explored. These events could play a crucial role in oncogenic signaling pathways. Lastly, did the authors consider the potential loss of *Pten* heterozygosity (LoH) through epigenetic mechanisms by performing tumor immunoblots?

scRNA-seq data display and quantitative analysis: The single-cell data are reported in aggregate, making it challenging to assess the consistency of specific phenotypes within a particular experimental group. A more quantitative approach would involve plotting the frequency of specific subpopulations by genotype and conducting statistical comparisons.

Concluding model:

The model provided in Figure 6 is not fully supported by the data.

- The authors have not demonstrated that MAPK kinase activation through mechanisms other than BRAF mutation leads to the expansion of LNM cells.
- While the study examines PTEN loss, it is not necessarily true that other homozygous tumor suppressor losses would promote the rapid onset of pigmented phenotypes. For instance, p53 or p16 loss in BRAF mutant mice results in rapid, amelanotic melanoma development (Viros et al., Nature 2015; Bowman LSA 2021).
- The impact of BRAF inhibition on LMN cell frequency is unexplored in this study.
- Amelanotic tumors can arise independently of *Sox2* (as shown in Figure 3D). Additionally, the role of *Sox2* in pigmented *BrafCA/+; PtenΔ/Δ* mice is not directly compared to *BrafCA/+; PtenWT/Δ* mice in this manuscript.

Data and methods reporting: The source and allele used to create albino mice are not provided, but this information is crucial for establishing that the models are truly congenic. Similarly, the strain and number of animals containing the mTmG reporter system should be reported for the single-cell analyses. A supplemental table showing sample type, genotype, strain, sample location (e.g., back, ear, tail, head), animal age, and animal sex for each of the 47 single-cell samples would be helpful.

Regarding the tumors and normal skin samples for WES and scRNA-seq, it would be valuable to clarify whether they were taken from the furred, dorsal area of the mouse versus the tail, ear, or paw, where melanocyte localization is known to differ.

Additionally, please provide an explanation of what each column represents in the figure legends for the heatmaps in Figures 2 and 3.

Reviewer #2

(Remarks to the Author)

Manuscript: NCOMMS-24-08299-T

Title: "Uncovering Minimal Pathways in Melanoma Initiation"

Xiao and colleagues have generated several genetically engineered mouse melanoma models by introducing inducible oncogenic/activated BRAF combined with a heterozygous loss of PTEN or wild-type PTEN on a Black and Albino background, followed by genomic analyses (WGS and scRNAseq profiling). Surprisingly, on an Albino background oncogenic BRAF activation alone seems sufficient to initiate melanomagenesis without significant recurrent genetic alterations as shown by whole-exome sequencing of three corresponding tumors. The authors reason that non-genetic transcriptional reprogramming might contribute to melanoma initiation. They identify, using single-cell RNA sequencing (345k cells from 36mice), a cell state termed LNM for Low-pigment, Neural-and extracellular Matrix, which presents in fast-growing murine tumors (persists transplantation), human biopsies and normal skin. Computational analyses predict LNM cells as source of other more abundant melanoma cell states during melanoma initiation and progression.

The proposed concept of non-genetic phenotype switching as a contributing mechanism to tumor initiation is intriguing. However, if the LNM melanoma cell state is the source, needs further investigation.

The following points should be addressed:

- The WGS analysis should include CCF estimation as well as mutational signature detection (even if the study is not using any UV treatment)
- What might be the cell(s) of origin in albino skin (BrafCA/+). Live-imaging possible? BRAFV600E specific antibody staining? see PMID: 29033351
- Are LNM1,2 Aqp1 positive melanoma cells also Sox2 positive? Could you please provide an Aqp1/Sox2 scatter plot? Aqp1 and Sox2 double staining? Especially the 3 BrafCA/+ Albino background tumor samples which were used for WGS?
- Aqp1 as a single marker for LNMs seems limited. Could you provide a higher resolution of the LNM cell state? Table of all LNM markers... Also, do all samples (tumor/skin of both backgrounds) contain LNM cells?
- Would it be possible to sort Aqp1 high cells from normal and malignant skin (Albino and Black)? To perform more precise molecular profiling. Aqp1 is also expressed in endothelial cells, btw.
- In general, the manuscript could benefit from CNV inference regarding malignant/pre-malignant/normal cell states or types (Honeybadger for example).
- The following paper should be discussed: Hodis et al. 2022 (PMID: 35482859) where they created nine genetically distinct models of melanoma by sequentially introducing mutation in healthy human melanocytes, using CRISPR-Cas9 genome editing. scRNAseq resolved states should be used for comparison with the presented study.
- Material and Methods: -scRNAseq analysis-
“cells with fewer than 4000 genes per cell” seems like a stringent cut-off. Normally, to exclude multipllets, fewer than 7500 genes per cell is acceptable when combined with Doubletfinder (or else) analysis.
- Figure 5E: the serially transplanted tumor #1 LNM cell population seems to split into clusters 0,1,2 with cluster 1 LNM cells majorly transitioning into cluster 3. What are the molecular differences between LNM clusters 0,1,2? What are the expression similarities between strong switchers: cluster 1 of tumor #1 (serially transplanted) and cluster 0 of tumor #2 (serially transplanted)?
- Regarding the “transitional cells” (Figure 5G), I recommend the authors to compare their data with Wouters et al. 2020 (PMID: 32753671), knowing that this is an in vitro data set but longitudinally resolved.
- Material and Methods: -Membership score analysis-
“variation of biological gene expression noise of 0.55, 0.65, 0.65 respectively” I don't get why two times 0.65?

Reviewer #3

(Remarks to the Author)

The manuscript by Xiao et. al. describes the kinetics of melanoma formation in two mouse models of melanoma. They also perform genetic profiling and single-cell RNA-sequencing of tumors that arise in these settings. Their models of melanoma are intriguing in that they have fewer driver mutations than most murine models of melanoma (namely Albino BRAFV600E or Black BRAFV600E PTEN-/+), which allows the authors to study the earliest steps in disease progression. The authors note that melanomas do form, albeit rarely, in these genetic backgrounds, and they suggest the melanomas arise via epigenetic (rather than genetic) transformations. They further study the epigenetic states of melanocytes, describing a new population of cells that they call LNM melanocytes, and relate these findings to human disease.

Minor points:

The notion that tumors arise from non-genetic events is virtually impossible to prove, as it relies on a negative result (the author's inability to find a cooperating mutational driver). This is not grounds to dismiss the idea, but the authors should acknowledge this limitation at the outset.

Given that the genetic background is important, maybe the BRAFV600E mutations are cooperating with a germline mutation. In this scenario, the germline mutations do not necessarily need to be engineered or known, but could have arisen over time through inbreeding of the mouse strains. Even if this were true, there is clearly an additional “event” occurring in the author's mice, given the stochastic, slow nature in which tumors arise, so this idea does not invalidate the main results of the paper. However, the proposition that a melanocyte with a single driver mutation can turn into a melanoma is radical, and given that this cannot happen in most strains of mice, it probably does depend on some underlying germline genetic hit.

Major points:

Instead of claiming that melanomas did not arise via loss of the second allele of PTEN, due to the pigmentation levels of the tumors, the authors should show at the genetic, transcriptional, or protein level that functional PTEN does exist.

The data supporting the conclusion of a genetic hit NOT driving tumor formation in their mice was not presented in a convincing fashion. The authors report a summary of mutations in figure 1D that arose in 3 tumors from the mice. The lack of recurrence is not evidence that there are no secondary hits, as each tumor could have acquired a unique mutation. They argue that these are passenger mutations, but it was difficult to review the mutations in table S1-3. The authors should annotate the coding mutations with HUGO gene names and protein changes using a standard output such as annovar or funcoator. As of now, they annotate genomic changes and transcriptomics changes linked to Ensembl IDs. The authors also claim that the allele frequencies of many variants are low, but this information is not available in the manuscript. Overall, it is challenging for a reader to validate the claim that there are no meaningful somatic alterations, aside from the ones engineered into the mouse models.

The authors claimed to have performed CNV analysis, but it appears that they only used a structural variant program to identify small insertions and deletions. The authors should also perform a coverage-based assessment of copy number, which would pick up on broad chromosomal changes at the kilo- to mega-base scale, which are difficult to detect with a structural variant caller. It would not surprise the reviewer if chromosomal arm gains/losses were present in the tumors. In the PTEN heterozygous mice, this could be a mechanism to lose the second copy of PTEN.

UMAPs can be deceiving, but why do the LNM cells appear to cluster closer to the melanocytes and Schwann cells in figure 2C? Despite the claims by the authors that LNM cells are closer to tumor cells?

In figure 4A Albino BrafCA⁺, it appears that the LNM cells cluster to the far left, and this cluster is exclusively skin cells, yet the authors state throughout the manuscript that LMN cells are enriched in tumors. More generally, figure 4 is challenging to follow.

Version 1:

Reviewer comments:

Reviewer #1

(Remarks to the Author)

Analysis of the comprehensive single cell dataset generated by Xiao et al. supports the emerging concept that non-mutational processes contribute to melanoma formation. While this analysis is timely and valuable, several concerns remain.

1. The mechanism by which the albino allele promotes tumor initiation and triggers LMN expansion is unknown.
2. The source, interchangeability, and relative significance of the LMN1 and LMN2 cell subsets is unclear.

Further clarification and appropriate modification of the authors' claims might mitigate these concerns.

Concerns

1. The manuscript concludes that pigment limits the onset of slow-growing melanomas, but the supporting data are based on a single genetic model (BRAF +/- Pten). To make this claim, the authors would need to determine why melanoma formation is enhanced in albino mice or rigorously explore an endless list of other possible explanations. I applaud the authors for making a commendable effort to address a subset of potential mechanisms in a few tumors. However, if identifying the mechanism of enhanced melanoma genesis is beyond the scope of this paper, the authors should avoid claims linking pigmentation or mutation-independent processes to tumor initiation and rapid growth.

2. It is not clear to me that the LMN2 population is a subset of the LMN population shown in Figure 2. This distinction is significant because these cell subsets are used interchangeably throughout the manuscript, and both are hypothesized to promote tumorigenesis. Specifically, the following concerns should be addressed:

- a. The relationship between LMN1 and 2 cells is not overly convincing, nor is it clear if the LMN2 subset is truly part of the original LMN subset defined in Figure 2. LMN1 and 2 cells are as different from one another as they are from tumor cells (Euclidean distance = 18.95 and 19.57, respectively). Why do the authors assume that LMN2 cells are a subset of LMN cells? How different are LMN2 cells from the other cell subsets depicted in Figure 2D (i.e. normal melanocytes, Schwann cells)?
- b. According to the velocity trajectories, LMN2 cells convert to LMN1 cells before becoming a tumor. If this is true, then why do LMN2 cells increase after serial transplantation? Where are the LMN2 cells coming from?
- c. I am not sure how the data in Figure 3D establish that Sox2 and Aqp1 are highly expressed in LMN1 and 2 cells. What is the comparator? Have the authors conducted any statistical analyses?
- d. As most LMN1, LMN2, and tumor cells do not express Sox 2 (Figure 3C), how do the authors propose that Sox2 deletion impacts tumor growth? Is it just that homozygous and heterozygous Sox2 knockout mice have fewer melanocytes to transform?

Minor Concerns

1. The homozygous/heterozygous status of the Tyr-CRE(ERT2) allele can influence recombination efficiency. Were all mice in the original study (Fig. 1B) homozygous for this allele? Please provide this information in the manuscript.
2. Please label the y-axis in Figure 3D.
3. The abstract and discussion require revision to eliminate unsupported generalizations regarding the pigmentation and mutational status of fast versus slow-growing melanomas.

Reviewer #2

(Remarks to the Author)

The authors have extensively improved their manuscript and addressed my points adequately. It is noteworthy that on an Albino background oncogenic BRAF activation alone seems sufficient to initiate melanomagenesis without significant recurrent genetic alterations (mutation or CNV). Malignant transformation seems to be tightly linked to the presence of a transcriptional cell state termed LNM for Low-pigment, Neural-and extracellular Matrix, which presents in fast-growing murine tumors (persists transplantation), human biopsies and normal skin. This is a significant finding and extends the knowledge about melanoma initiation. For example, Köhler et al. showed that mouse cutaneous melanoma induced by mutant Braf arises from the expansion and dedifferentiation of mature pigmented melanocytes (PMID: 29033351). I believe that the manuscript uncovers another important piece of the "transcriptional plasticity puzzle" in melanoma and recommend its publication.

Reviewer #3

(Remarks to the Author)

I thank the authors for their increased transparency and improved reporting in this revision. Unfortunately, this revealed major issues with the conclusion that tumors are arising via non-genetic events.

In tables S1 and S3, the authors detect Flna mutations. In the main text, they claimed "no genes were mutated across all three tumors", but this is an arbitrary threshold, especially because there is a gene, Flna, that is recurrently mutated in two out of three tumors. Of note, the Flna mutations have the highest allele frequencies of any coding mutations in these tumors, arguing that they are clonal. Initially, this reviewer was unsure what to make of Flna mutations, however, after searching the literature, mutations in this gene are VERY clearly under selection in mouse melanomas. A manuscript by Bowman and colleagues (Life Science Alliance, 2021) from the laboratory of Dr. Christin Burd found mutations in Flna in 42% of murine melanomas that developed. While there remain questions as to how Flna mutations drive melanoma, given the high frequency of mutations in mouse melanoma, the dN/dS ratio of mutations in mouse melanomas, and now the discovery of recurrent mutations in two independent studies, this gene is clearly a driver gene.

It was serendipitous that the reviewer discovered the manuscript by Bowman and colleagues because there were other genes and gene families mutated in that study that had mutations reported here in tables S1, S3, and S5. For example, Vmn and Gm genes were seen in both studies. There is a strong chance that additional driver mutations, beyond Flna, are present but were dismissed due to the unjustifiable requirement that they be present in 100% of tumors.

While the authors have improved their reporting of point mutations, the reporting of copy number alterations remains subpar. Supplementary tables 7-12 present lists of breakpoints and copy number segments from breakdancer and CNVkit. A human cannot easily interpret these tables – these outputs were intended to be read by scripts. A better way to present these results would be circos plots (for structural variants) and scatterplots of probe- and segment- level copy number inferences (for copy number changes). As a point of reference, the authors should show copy number data similar to figure 5 of the original CNVkit paper (<https://journals.plos.org/ploscompbiol/article/figure?id=10.1371/journal.pcbi.1004873.g005>).

The authors do show copy number inferences from inferCNV in supplementary figure 4. In tumor 3, there is clearly a gain on chromosome 6. In tumor 2, there may also be a gain on chromosome 6. These gains are notable because the Braf gene is on chromosome 6.

In conclusion, two tumors have Flna mutations and a third tumor has a broad copy number gain which encompasses Braf. There are other candidate drivers among point mutations, and an improved presentation of copy number data may yet reveal more alterations driving these tumors. The authors describe some interesting non-genetic changes that occur during the early stages of melanoma, but the idea that tumors are arising purely from non-genetic changes does not stand scrutiny.

Version 2:

Reviewer comments:

Reviewer #1

(Remarks to the Author)

The authors have addressed most of my previous concerns. The mechanism may be beyond the scope of this publication, but several issues remain regarding the claim that the albino allele leads to rapid tumor growth.

The copy number of Tyr:CreERT(2) may influence the rate of tumorigenesis. The authors state that all study mice were heterozygous for Tyr:CreERT(2). However, the provided genotyping primers only amplify Cre, and the breeding scheme does not preclude the birth of homozygous animals. The authors should use a primer set specific to the Tyr:CreERT(2) insertion site (see: Aktary et al.; PMID: 31679174) to confirm the zygosity of the experimental mice. Once confirmed, these data should be included in the Results section, and both breeding schemes shown in Supplementary Figure 1.

Were the pups from the second cross in Supplementary Figure 1C (left panel, right side of the cross) separated before tamoxifen treatment to ensure those used for subsequent breeding were not exposed to tamoxifen? Although these animals would constitute only a small portion of the study, it is important to examine whether prior tamoxifen exposure did enhance tumorigenesis in the offspring.

There is insufficient evidence to show that the loss of pigment, rather than Tyrosinase function, leads to changes in melanoma development. Therefore, it is suggested that the authors:

- Remove the claim about pigmentation in the following sentence (lines 101-103): “Nonetheless, it is clear that pigmentation status and Pten gene dosage both influence the probability of melanoma initiation in *BrafCa/+* and *BrafCa/+ PtenD/+* mice.”
- Replace “albino” with “Tyrosinase-deficient” in the statement on lines 371-373.
- Consider revising the model in Figure 6, replacing “albino” with “Tyr C-2J homozygous.”

Reviewer #3

(Remarks to the Author)

I believe the in-frame indel in *Flna* of tumor 2 is a driver mutation. In argument against this position, the authors begin by mentioning that *Flna* mutations are rare in human melanoma, but this is irrelevant because they are drivers in mouse melanoma. Next, the authors note that this *Flna* mutation has an allele frequency that is 1.6X the inferred tumor cell content. The high allele frequency of the in-frame indel could have easily arisen from random over sampling of the mutant allele in the sequencing data, and/or the authors estimate of tumor cellularity could be too low. The authors next point out that the in-frame indel has not yet been observed in other mouse melanomas. While the exact (or similar) indel mutation in *Flna* might not have been observed in another tumor (as of yet), it is extremely common for cancer genes to have minor hotspots. As an example, one can look at the distribution of BRAF mutations in human melanomas – V600E is most common, but there are plenty of other hotspots, mutated at less than 5% frequency. The mutation burden of the mouse tumors in this study is low and considering this, it is difficult to fathom how a passenger mutation arose in a driver gene. Taken together, this reviewer believes that *Flna* driver mutations are present in 2 out of the 3 tumors that were deeply characterized by the authors, and apparently they are in additional tumors with single-cell RNA sequencing data, which is telling because scRNA-seq is not a sensitive method to detect mutations.

While this reviewer does not recognize an obvious driver mutation in the third tumor or additional driver mutations beyond the *Flna* mutation in the first two tumors, there is quite a lot of noise in every genetic result in the manuscript. For example, the authors point out that their point mutations are potentially full of artifacts in *Vmn* and *Gm* genes, and there are hundreds of structural variants and copy number breakpoints. Given this level of noise, I do not feel comfortable with claims that there are no additional driver mutations.

Overall, I do agree when the reviewers say “our proposal that non-genetic processes may be required for tumor initiation remains worth considering”. The authors insinuate that non-genetic processes may be sufficient for tumor initiation – the repetitive claims that they failed to find driver mutations in a majority of tumors certainly lends itself to this interpretation. My take is that genetic processes are occurring alongside non-genetic processes to drive the evolution of the melanomas in this study. I am not convinced that these tumors are arising in the absence of genetic drivers.

Response to Reviewers: Reviewer comments are highlighted in **blue** while author responses are in **black**.

Reviewer #1 (Remarks to the Author): Expert in melanoma development, genetically engineered mouse models, molecular and cellular biology

Understanding the genetic events that drive melanoma progression has proven challenging due to the high mutational burden observed even in normal skin. Additionally, the presence of BRAF and NRAS mutant melanocytes in benign human nevi shows that MAPK activation is necessary but insufficient to stimulate melanoma progression. Here, Xiao et al. use single cell sequencing to analyze tumor progression in genetically engineered mouse models of spontaneous Braf-mutant melanoma. They identify a distinct cell population characterized by low-pigment, neural, and extracellular matrix (LNM) gene expression. These LNM cells exist in normal skin, expand upon BRAF activation, and persist through multiple passages in mice. Notably, the transcriptional profile of LNM cells closely resembles neural crest-like populations seen during the dynamic, phenotype switching of human and murine melanoma cells. These findings suggest that phenotype switching plays a pivotal role not only in melanoma progression but also in its initiation.

Beyond its implications for melanoma development, this study underscores the significance of non- mutational, epigenomic events in melanoma initiation. Further, the comprehensive single cell sequencing dataset, comprising over 45 samples from one of the most used pre-clinical GEMMs of melanoma, will undoubtedly serve as a valuable resource for the research community.

We thank the reviewer for recognizing the importance of the work and its utility as a resource to the melanoma community. A point-by-point response to specific concerns is presented below.

Several limitations must be addressed to strengthen the scientific rigor, conclusions, and significance of this work. These limitations, include efforts to: 1) determine how the albino allele promotes tumor initiation in specific genetic contexts, and clarify its relevance to fair-skinned individuals who produce pheomelanin—a factor associated with carcinogenesis, 2) clearly define and consistently analyze the LMN1 and LMN2 cell populations as separate entities, 3) refrain from, conclusions based on comparisons with genetically distinct mouse models conducted by other groups, 4) improve transparency, rigor, and accuracy by further refining the analytical methods, data reporting, and the final model.

Specific Concerns

Tumor promotion by the albino allele: The authors suggest that albinism contributes to melanoma progression in mice. They suggest that this is similar to evidence that skin color affects human melanoma risk independently of mutations. However, even individuals with light skin tones still produce melanin. Furthermore, in their Pten heterozygous model, albino mice exhibited slower tumor development (Figure 1B). How the albino allele impacts tumor progression in a genotype-dependent manner is never addressed.

We acknowledge that there is a distinction between animal models that make some melanin, or different types of melanin, and those animal models that make no melanin. Of note, *TYR* mutations are associated with melanoma risk¹, and albino patients can develop melanomas², so melanomas can develop with little or no pigment. The utility of the models described here is not meant to be a system to study albinism in melanoma but rather as a model to understand how *Braf* mutation alone can lead to melanoma and identify the epigenetic transitions that lead to melanoma development. The human relevance of the identified transitions is illustrated by the

fact that the populations observed here are conserved in human tumors. We discuss this in lines 321-329 of the discussion.

The reviewer is also correct that *Pten* heterozygous albino mice developed tumors at a slower rate than *Pten* heterozygous black mice. We now mention in the discussion about how the albino allele could have either positive or negative effects on tumor development, depending on the genetic background (lines 316-320). Elucidating the differences in mechanism is beyond the scope of the current study but will be a focus of future work.

Analysis of the LMN1 and LMN2: The authors' conclusion that LMN cells increase through successive tumor transplantations raises several questions. Notably, LMN1 cells appear to decrease in frequency with subsequent transplantation whereas, LMN2 cells increase. How the LMN2 population was determined to be LMN1-like is unclear – especially since LMN2 cells lack several key markers defined in the original analysis (e.g., *Aqp1*, *Igf1*, *Sema3b*; Figure 3C).

One limitation of the data as initially presented was that it was difficult to delineate the differences between LNM1 and LNM2 due to the limited numbers of cells in the transplant experiments. To improve the overall data analysis robustness, we added another set of transplantation experiments (different donor genotype, same NSG host genotype, but with only one round of transplant). We then re-analyzed all the transplanted data using Harmony, which has the improved ability to cluster cells generated in the context of different genetic backgrounds (in this case NSG versus B6). Figure 3C now shows the markers shared between LNM1 and LNM2, including *Aqp1* and *Sox2*. We specifically examine the co-expression of *Sox2* and *Aqp1* in LNM1 and LNM2 populations (Figure 3D). We have also included the differentially expressed gene lists for clusters of NC-derived cells (Fig 2C) in Supplementary tables 15 and 16.

It is difficult to read the Euclidean distance in Supplemental Figure 4C, but it appears that the LNM1 and LMN2 cells are almost equally different from one another as they are from the tumor or melanocyte populations.

We have replaced the diagrams in the initial version with Euclidean distance heatmaps that should be easier to follow (Fig S6D). These heatmaps show that the distance is closest between Tumor and LNM1 (9.16), and about equivalent between LNM2 vs. Tumor (19.57) and LNM2 vs. LNM1 (18.95).

In the tumor cell projections (Figure 5), sometimes the LMN populations are divided into LMN1 and LMN2 and sometimes they are not. This seems like an important designation given that LMN1 frequently appears to decrease whereas LMN2 numbers increase with tumor transplantation.

We apologize for this confusion. Due to the complexity of the analysis here, we had to analyze tumors and transplants individually. When we did so, LNM cells separated into two clusters in the context of tumor transplantation, while in parental tumors they did not separate into two clusters. We have done a better job listing both genes expressed by LNM1 and LNM2. These clusters are related as LNM2 expresses many of the same genes as LNM1 to a lesser extent (Fig 3C).

Finally, *Sox2* appears to be most strongly expressed in LMN1 cells which are lost during multiple rounds of tumor grafting (Fig. 3C). This brings into question the relevance of *Sox2* in LMN-dependent tumor progression.

In the revised manuscript, we have clarified that *Sox2* and *Aqp1* co-expressing cells are present in the LNM1 and LNM2 populations (Fig 3C, 3D). We have also clarified that LNM1 cells, while

a small population, are not lost during subsequent rounds of tumor grafting (Fig S6C). Their gene expression pattern, in fact, is more closely related to tumor cells (Fig S6D).

Internal animal controls: Due to potential confounding factors such as genetic drift, methodological variations, and animal husbandry practices, comparisons between mouse models should be conducted under identical conditions—especially given their significance to the authors' model. Examples of specific concerns related to this technical limitation are:

In Figure 1C (lower right), only one image documents the pigmented state of *Braf*^{CA/+}; *Pten*^{Δ/Δ} mice. It appears to show intact hair follicles. Did a pathologist confirm that this was a bona fide tumor and not a melanocytic hyperproliferation, which is known to occur in the *Braf*^{CA/+}; *Pten*^{Δ/Δ} model?

We thank the reviewer for picking this up. This was the result of not showing a representative section. We have had all the *Braf*^{CA/+}; *Pten*^{Δ/Δ} tumors analyzed by a pathologist, Robert Edwards, who verified they were tumors, and is now an author on the paper. Figure 1C has been amended to include a representative tumor.

The authors propose that *Sox2* deletion limits tumor growth in *Braf*^{CA/+}; *Pten*^{Δ/+} mice but not in *Braf*^{CA/+}; *Pten*^{Δ/Δ} animals. However, data for the *Pten* null phenotype is not shown to validate replicability of published studies in their lab. Additionally, tumor number may not reflect onset, but differences in aggressive behavior of the melanomas that impact euthanasia timing. Were there variations in the Kaplan- Meier curve or tumor growth rates of *Sox2* null and wildtype mice?

We agree with the reviewer that we should refrain from comparing results between labs. Instead of directly comparing the results, we have changed the wording to acknowledge prior published work (lines 237-239) without comparing the results to the current work (lines 348-353). With regards to the growth behavior, we only observed one tumor in the *Braf*^{CA/+}; *Pten*^{Δ/+} *Sox2*^{Δ/Δ}, making any such comparisons of the growth rate not possible.

Data transparency, rigor, and accuracy

Variant calling: The number of variants detected in this study exceeds what has been reported for similar spontaneous melanoma models in the literature. Notably, many of these changes are observed in genes with fewer than five reads, as indicated in Supplemental Tables 1-6. To address this, it's essential to understand how the minimum threshold for mutation and copy number variant (CNV) calling was determined.

Additionally, please provide a key for the headers in Supplementary Table 1, along with clarification on how each normal sample was used during the variant calling process.

We have included more detailed Annovar output tables that report the allele frequencies for all detected mutations. We have presented the counts of each SNV/CNV and the overall coverage rather than presenting selected SNVs that exceeded a threshold so readers have access to all of the data. We have also amended table 1D to report the number of exonic mutations so one can compare with existing datasets. For SNV analysis (mutect2) and CNV analysis (BreakDancer and CNVkit), default parameters were used. These settings were selected based on their widespread validation and suitability for our specific analytical requirements. Spleen and Skin tissues were used as normal samples. A comprehensive description of the analysis process, including the role of the normal samples, is thoroughly detailed in the Methods section of our manuscript.

Regarding the comparisons of our results with those of others, expectations depend on how deep one sequences, whether there is a numerical threshold for calling a mutation, and whether the

whole genome, or only exons, are sequenced. To our knowledge, most previous work with which we can compare focused on exonic sequences, for which reported mutation prevalences are not dramatically different from what we report here. For example, in a study by Viros et al., (2014) numbers of exonic mutations in melanomas from *Braf* mutant, non-UV irradiated mice ranged 1 to 103 per tumor, with median = 9 and mean =18³ (these tumors used a slightly different *Braf* mutant allele from ours). In our study, we observed 25 to 110 exonic mutations (median = 27 and mean =54). The difference possibly reflects lower sequencing depth in the Viros study, as well as their use of filters to remove “likely” false positives (whereas, in the interest of not missing any true positives, we called all variants supported by as few as two reads).

Regarding the CNV analysis, it's intriguing that gene amplifications were not explored. These events could play a crucial role in oncogenic signaling pathways. Lastly, did the authors consider the potential loss of Pten heterozygosity (LoH) through epigenetic mechanisms by performing tumor immunoblots?

For the three Albino *Braf*^{CA/+} tumors subjected to WGS, we supplemented our previous analysis by Breakdancer (which pinpoints CNVs based on their breakpoints⁴), with a coverage-based analysis of copy number variation using the CNVkit algorithm . Finally, we also used inferCNV to infer CNVs from transcript expression data.

CNVkit attempts to detect structural variants based on differences in sequence coverage when compared with a reference sample (the lower the percent tumor cells, the smaller a difference in coverage needs to be in order to be considered consistent with a structural variant). CNVkit (and other read based CNV detection algorithms for that matter⁵), is rather prone to making false positive calls when used on WGS, especially for indels <1 Mb. In our hands it is particularly likely to mis-call copy number for genes with close homologues (or pseudogenes) elsewhere in the genome (presumably due to mismapping). To get a sense of where to set the detection thresholds so as to control false positivity to a reasonable level, we compared the spleen (control) DNA from each of the three tumor animals against the spleens of the other two (we would not expect to see a large number of high-frequency structural variants in spleen DNA). Using a detection threshold that limited spleen calls on autosomes to less than 101 per animal, CNVkit called out between 3.5 and 13.6 times this number in the tumor samples, of which 55 potential variants were shared among all tumors, although 10 of these were obvious examples of highly homologous gene groups. Of the 55, only four (all marked as heterozygous deletion) overlapped with any CNVs identified by BreakDancer in any tumor. None of these four were identified by BreakDancer in more than one tumor, and their observed breakpoints were grossly inconsistent in size with the regions suggested to have been altered by CNVkit; Furthermore, except in one case, the calls by BreakDancer for these genes were supported by very few reads (2-3) so that, even if they are real they are unlikely to have been founder events in tumor formation. In addition, none of these CNV regions were identified when we used inferCNV to infer CNVs from single cell gene expression data. We present all the putative CNVs identified by Breakdancer and CNVkit as a resource for the reader in case they have interest in any of the particular genes/regions amplified (Supplementary Tables 7-12). This is now discussed in lines 133-142 and 476-491 of the manuscript.

On balance, we believe the data do not provide evidence of any consistent mutation or structural variant in albino *Braf*^{CA} tumors, and that the most parsimonious explanation is that tumor initiation does not require such a change. We recognize that no structural variant detection method is perfect, and the possibility that a common causal variant might be detected in future cannot be eliminated, nor the possibility that different variants or mutations are acting in different tumors.

We have also performed immunofluorescence staining of *Pten* heterozygous and wild type tumors and confirmed that both tumor types expressed *Pten*, confirming that the phenotype was not a result of loss of heterozygosity. The results are shown in Figure S2A.

scRNA-seq data display and quantitative analysis: The single-cell data are reported in aggregate, making it challenging to assess the consistency of specific phenotypes within a particular experimental group. A more quantitative approach would involve plotting the frequency of specific subpopulations by genotype and conducting statistical comparisons.

Direct statistical comparisons are difficult due to the differing number of cells sequenced in each animal. Thus, we have addressed this in a slightly different way by annotating the single cell data by sample and showing the data are well mixed (Fig S3A, S3C, S6B, and S7A). This shows the consistency of the phenotypes across samples. We also provide more quantitative comparisons of percentages of each type after transplant (Fig S6C and Fig S7C).

Concluding model:

The model provided in Figure 6 is not fully supported by the data.

- The authors have not demonstrated that MAPK kinase activation through mechanisms other than BRAF mutation leads to the expansion of LNM cells.
- While the study examines *PTEN* loss, it is not necessarily true that other homozygous tumor suppressor losses would promote the rapid onset of pigmented phenotypes. For instance, *p53* or *p16* loss in BRAF mutant mice results in rapid, amelanotic melanoma development (Viros et al., Nature 2015; Bowman LSA 2021).
- The impact of BRAF inhibition on LMN cell frequency is unexplored in this study.
- Amelanotic tumors can arise independently of *Sox2* (as shown in Figure 3D). Additionally, the role of *Sox2* in pigmented *Braf^{CA/+}; Pten Δ/Δ* mice is not directly compared to *Braf^{CA/+}; Pten^{WT/\Delta}* mice in this manuscript.

We thank the reviewer for this comment, particularly regarding *MAPK* activation and effects of the inhibitor. We have focused only on *Pten* loss or partial loss of *Pten* in our model (as suggested by the reviewer) in an effort to contextualize our findings, but have refrained from directly comparing the *Sox2* null tumor results across models.

Data and methods reporting: The source and allele used to create albino mice are not provided, but this information is crucial for establishing that the models are truly congenic. Similarly, the strain and number of animals containing the mTmG reporter system should be reported for the single-cell analyses. A supplemental table showing sample type, genotype, strain, sample location (e.g., back, ear, tail, head), animal age, and animal sex for each of the 47 single-cell samples would be helpful.

We have now described the strain and included a pedigree chart describing the generation of all albino *Braf^{CA/+}* mouse models in Fig S1C. As we now note in the methods, based on this scheme, if an unlinked homozygous modifier had been present in both founder animals, the probability of it appearing in homozygous form in all tumor-bearing mice is less than 1/10,000, and the probability of it appearing in heterozygous form <7.5%. The mouse metadata information is now included as a table in Supplementary table 13; this also includes the number of mTmG animals.

Regarding the tumors and normal skin samples for WES and scRNA-seq, it would be valuable to clarify whether they were taken from the furred, dorsal area of the mouse versus the tail, ear, or paw, where melanocyte localization is known to differ.

All samples (skin and tumor) were collected from the dorsal back skin. This is now clarified in the metadata table in Supplementary table 13.

Additionally, please provide an explanation of what each column represents in the figure legends for the heatmaps in Figures 2 and 3.

Each column represents an individual cell from the corresponding group (in this case, cell type). The heatmap was downsampled for readability. This information has now been provided in the legends.

Reviewer #2 (Remarks to the Author): Expert in single-cell RNAseq, melanoma genomics, and computational genomics

Manuscript: NCOMMS-24-08299-T

Title: "Uncovering Minimal Pathways in Melanoma Initiation"

Xiao and colleagues have generated several genetically engineered mouse melanoma models by introducing inducible oncogenic/activated BRAF combined with a heterozygous loss of PTEN or wild-type PTEN on a Black and Albino background, followed by genomic analyses (WGS and scRNAseq profiling). Surprisingly, on an Albino background oncogenic BRAF activation alone seems sufficient to initiate melanomagenesis without significant recurrent genetic alterations as shown by whole-exome sequencing of three corresponding tumors. The authors reason that non-genetic transcriptional reprogramming might contribute to melanoma initiation. They identify, using single-cell RNA sequencing (345k cells from 36mice), a cell state termed LNM for Low-pigment, Neural-and extracellular Matrix, which presents in fast-growing murine tumors (persists transplantation), human biopsies and normal skin. Computational analyses predict LNM cells as source of other more abundant melanoma cell states during melanoma initiation and progression.

The proposed concept of non-genetic phenotype switching as a contributing mechanism to tumor initiation is intriguing. However, if the LNM melanoma cell state is the source, needs further investigation.

We appreciate that the reviewer recognizes the intriguing mechanism proposed to explain melanoma initiation and below we address their concerns regarding the mechanism.

The following points should be addressed:

The WGS analysis should include CCF estimation as well as mutational signature detection (even if the study is not using any UV treatment)

One benefit of the approach here is that we can more directly measure the tumor cell fraction by quantifying the frequency of recombination of the *Braf* locus. We have described this approach in the methods and the estimated fraction of cancer cells in each tumor (lines 108-110). We also report the allele frequency (Supplementary table 1, 3, and 5) for each mutation and detect the mutation signature using sigProfiler. A graph representing the number of single base substitutions in the transcribed and untranscribed strand is provided (Fig S2B) as requested.

What might be the cell(s) of origin in albino skin (*Braf*^{CA/+}). Live-imaging possible? BRAFV600E specific antibody staining? see PMID: 29033351

The reviewer asks a very important and astute question, but it is one we can't answer at the current time. The challenge is the rarity of tumors in the Albino *Braf*^{CA/+} mice (only one tumor per animal on average) which makes it nearly impossible to track from where these tumors start. Ongoing work seeks to identify markers of transitioning cells, which we hope will help track them. But that is beyond the scope of the current work and not feasible at the moment.

Are LNM1,2 Aqp1 positive melanoma cells also Sox2 positive? Could you please provide an Aqp1/Sox2 scatter plot? Aqp1 and Sox2 double staining? Especially the 3 *Braf*^{CA/+} Albino background tumor samples which were used for WGS?

We thank the reviewer for this comment. Given the considerable sparsity of scRNA-seq data, proving a statistically significant linear relationship between any two genes across individual cells is quite challenging. To better demonstrate their co-expression, we have now provided a Violin

plot to show a consistent expression pattern of *Aqp1* and *Sox2* in both the LNM1 and LNM2 clusters (Fig 3D). We have also now included differentially expressed gene lists to better define markers of the LNM1 and LNM2 populations (Supplementary table 16). We have also performed co-staining of the *Aqp1* and *Sox2* on two of the Albino *Braf*^{CA/+} tumor samples used for WGS and used RNAscope (Fig S6E) to show that some of the cells in the superficial dermis express both markers.

Aqp1 as a single marker for LNMs seems limited. Could you provide a higher resolution of the LNM cell state? Table of all LNM markers... Also, do all samples (tumor/skin of both backgrounds) contain LNM cells?

We have included tables of the LNM markers as suggested, in Supplementary table 15-16. Both tumor and skin contain LNM cells, but the number of cells in normal skin is exceedingly low. This is now commented on in the manuscript, while we focus more on comparing the number of LNM cells in *Braf* mutant skin (Figure S3E) and tumors (Fig S6C).

Would it be possible to sort Aqp1 high cells from normal and malignant skin (Albino and Black)? To perform more precise molecular profiling. Aqp1 is also expressed in endothelial cells, btw.

As noted above, the frequency of LNM cells in normal skin is exceedingly rare and it is rare in tumors too, making it a challenge to get enough cells to sort and profile, and we feel these studies are beyond the scope of the current work. We do show that the *Aqp1* LNM population can be distinguished from *Aqp1*+ endothelial cells of blood vessels, based on location of staining (Figure S3G and S7C).

In general, the manuscript could benefit from CNV inference regarding malignant/pre-malignant/normal cell states or types (Honeybadger for example).

We now report results using the inferCNV pipeline for three Albino *Braf*^{CA/+} tumors, with fibroblasts as the normal cell reference, and using melanocytes, LNM cells, principal tumor cells, macrophages, endothelial cells, erythrocytes, and T cells as the observational cells. This pipeline was developed to use scRNAseq data to identify common CNVs between cell population and track those that might be causative of tumorigenesis. Because it is based on gene expression, it can give false positive results when cell type-specific gene expression (or lack of expression) happens to involve multiple genes that are located nearby on a chromosome. Adjustable parameters allow one to minimize false positives, but since we wanted to minimize false negatives, we used a generous FDR cutoff of 20%. At that level, one observes certain regions being called out in normal cells (e.g. macrophages) more or less identically in all samples—likely false positive calls (Fig. S4). Yet even at that level, we don't see any common or overlapping CNVs across the primary tumor cells from the three tumors.

The following paper should be discussed: Hodis et al. 2022 (PMID: 35482859) where they created nine genetically distinct models of melanoma by sequentially introducing mutation in healthy human melanocytes, using CRISPR-Cas9 genome editing. scRNAseq resolved states should be used for comparison with the presented study.

We thank the reviewer for this suggestion. We have now compared both the in-vivo and in-vitro states in the Hodis et al study, with those in our study. This is included in Figure S7E-G. Briefly, our principal tumor cells had features resembling the “EMT” signature of their CRISPR-edited in-vitro and in-vivo tumor cells. Our LNM cells most closely resemble the “Interferon/TGF β ” signature in their in-vivo CRISPR-edited models. And our *Mitf*-high melanocytes, are most aligned with their “melanocyte”, “Interferon/p53”, and “Myc/mTORC1/OxPhos” signatures in the CRISPR-

edited in-vitro cells; and the “melanocytic”, “OxPhos”, and “ β -catenin/MITF” signatures in their CRISPR-edited in vivo cells. This is discussed in lines 263-283.

Material and Methods: -scRNAseq analysis-

“cells with fewer than 4000 genes per cell” seems like a stringent cut-off. Normally, to exclude multiplets, fewer than 7500 genes per cell is acceptable when combined with Doubletfinder (or else) analysis.

We thank the reviewer for this comment. We examined the frequency of cells with 7500 and 4000 genes per cell. Based on the 4000 cutoff, we eliminated only 7.1% of the total cells. Therefore, it is unlikely that we eliminated too many cells that would affect clustering or affect the reported results.

Figure 5E: the serially transplanted tumor #1 LNM cell population seems to split into clusters 0,1,2 with cluster 1 LNM cells majorly transitioning into cluster 3. What are the molecular differences between LNM clusters 0,1,2? What are the expression similarities between strong switchers: cluster 1 of tumor #1 (serially transplanted) and cluster 0 of tumor #2 (serially transplanted)?

We have provided a more detailed table of the gene expression profiles of the LNM clusters from MuTrans to address this question. This information is now included in Supplementary table 22. Briefly, the clusters in Fig 5E were determined by the entropy level and attractor assignment from MuTrans, a pipeline to examine transition states from scRNA-seq profiles. Transplanted tumor #1 has more subsets of LNM cells, suggesting that this tumor has more transient cell states, though some LNM subset (in this case, cluster 1) might have more precursor signature than the others. (as shown in the dynamic manifold and probability path in Fig 5G-H). As shown in Supplementary table 22, the two strong switchers (cluster 1 of serially transplanted tumor #1 vs. cluster 0 of serially transplanted tumor #2) share markers of the meta (stable) states (e.g. *Timp1*, *Ranbp1*, *Cpe*, *Cstb*), and also their hybrid (transient) states (*Mt2*, *Esm1*, *Dut*, *Stab1*, etc).

Regarding the “transitional cells” (Figure 5G), I recommend the authors to compare their data with Wouters et al. 2020 (PMID: 32753671), knowing that this is an in vitro data set but longitudinally resolved.

We have now compared our dataset with that from Wouters et al and have included the results in Fig S7E-G. Briefly, our principal tumor cells had features resembling the “Mesenchymal-like” signature in their melanoma cultures. Our LNM cells most closely resemble the “Intermediate” signature in their data. And our *Mitf*-high melanocytes, are most aligned with their “melanocytic” signature. This is discussed in lines 263-283.

Material and Methods: -Membership score analysis-

“variation of biological gene expression noise of 0.55, 0.65, 0.65 respectively” I don’t get why two times 0.65?

As discussed in the Methods, the input to membership score analysis is a matrix of modified, corrected Pearson residuals, which provides a more accurate, principled way of representing cell-specific gene expression without the distorting effects of data normalization. This is described in the cited paper, which is both on biorxiv and now in press at BMC Bioinformatics, to appear shortly. Calculation of modified corrected Pearson residuals involves the use of a hyperparameter learned from the data, which quantifies the apparent coefficient of variation of underlying gene expression noise. The values 0.55, 0.65 and 0.65 correspond to the parameter values that were learned directly from the data. The differences likely reflect batch-specific variation in scRNAseq data, requiring slightly different degrees of correction in different samples.

Reviewer #3 (Remarks to the Author): Expert in melanoma genomics, origin and progression; and single-cell RNA-seq

The manuscript by Xiao et. al. describes the kinetics of melanoma formation in two mouse models of melanoma. They also perform genetic profiling and single-cell RNA-sequencing of tumors that arise in these settings. Their models of melanoma are intriguing in that they have fewer driver mutations than most murine models of melanoma (namely Albino BRAFV600E or Black BRAFV600E PTEN-/+), which allows the authors to study the earliest steps in disease progression. The authors note that melanomas do form, albeit rarely, in these genetic backgrounds, and they suggest the melanomas arise via epigenetic (rather than genetic) transformations. They further study the epigenetic states of melanocytes, describing a new population of cells that they call LNM melanocytes, and relate these findings to human disease. We thank the reviewer for recognizing the overall novelty of the work

Minor points:

The notion that tumors arise from non-genetic events is virtually impossible to prove, as it relies on a negative result (the author's inability to find a cooperating mutational driver). This is not grounds to dismiss the idea, but the authors should acknowledge this limitation at the outset.

We agree with the reviewer that we cannot absolutely prove that the tumors exclusively arise from a non-genetic event. We acknowledge that up front (lines 137-147) and report the frequencies and location of every mutation and putative CNV observed (Supplementary table 1-12, Fig 1D-E).

Given that the genetic background is important, maybe the BRAFV600E mutations are cooperating with a germline mutation. In this scenario, the germline mutations do not necessarily need to be engineered or known, but could have arisen over time through inbreeding of the mouse strains. Even if this were true, there is clearly an additional "event" occurring in the author's mice, given the stochastic, slow nature in which tumors arise, so this idea does not invalidate the main results of the paper. However, the proposition that a melanocyte with a single driver mutation can turn into a melanoma is radical, and given that this cannot happen in most strains of mice, it probably does depend on some underlying germline genetic hit.

The reviewer raises the concern that the albino mice might harbor a germline modifier mutation that could be responsible for the increased risk of melanoma development. As described above, given the breeding that was done, we calculate that, if an unlinked homozygous modifier had been present in both founder animals, the probability of it appearing in homozygous form in all tumor bearing mice was less than 1/10,000, and the probability of it appearing in heterozygous form <7.5%. (Fig S1C). Furthermore, mutations in tyrosinase are known to increase human melanoma risk, so the idea they do so in mice is not at all far-fetched. These issues are discussed in the revised manuscript (lines 321-329). The reviewer also comments that, since black *Braf*^{CA/+} mice display zero tumors and Albino *Braf*^{CA/+} mice display some, that there must be something categorically different going in the albino mice, such as an additional, unknown mutation (besides albinism). But observing zero tumors in black *Braf* mice doesn't mean tumor formation is impossible, just that it is so rare we don't detect tumors in relatively small samples of mice. As we point out in the manuscript, other groups who use a slightly different *Braf* construct do occasionally observe tumors in black mice⁶. What we envision is that the probability of tumor initiation is simply very low in Black *Braf*^{CA/+} mice, somewhat higher in Albino *Braf*^{CA/+} mice, and higher still in Black *Braf*^{CA/+} *Pten*^{Δ/+} mice. Of

course this does not tell us why such differences occur, but we don't think there is any compelling reason to propose that there must be unseen germline hits that are responsible.

Major points:

Instead of claiming that melanomas did not arise via loss of the second allele of PTEN, due to the pigmentation levels of the tumors, the authors should show at the genetic, transcriptional, or protein level that functional PTEN does exist.

We have now performed immunofluorescence staining of *Pten* heterozygous and wild type tumors and confirmed that both tumor types expressed *Pten*, indicating that the phenotype was not a result of loss of heterozygosity. The results are shown in Figure S2A.

The data supporting the conclusion of a genetic hit NOT driving tumor formation in their mice was not presented in a convincing fashion. The authors report a summary of mutations in figure 1D that arose in 3 tumors from the mice. The lack of recurrence is not evidence that there are no secondary hits, as each tumor could have acquired a unique mutation. They argue that these are passenger mutations, but it was difficult to review the mutations in table S1-3. The authors should annotate the coding mutations with HUGO gene names and protein changes using a standard output such as annovar or functotator. As of now, they annotate genomic changes and transcriptomics changes linked to Ensembl IDs. The authors also claim that the allele frequencies of many variants are low, but this information is not available in the manuscript. Overall, it is challenging for a reader to validate the claim that there are no meaningful somatic alterations, aside from the ones engineered into the mouse models.

We thank the reviewer for this comment. We have now revised the format of these tables to align with standard conventions in the field. The updated tables (Supplementary Tables 1-6) now include detailed headers and allele frequencies and are formatted using ANNOVAR as requested.

The authors claimed to have performed CNV analysis, but it appears that they only used a structural variant program to identify small insertions and deletions. The authors should also perform a coverage-based assessment of copy number, which would pick up on broad chromosomal changes at the kilo- to mega-base scale, which are difficult to detect with a structural variant caller. It would not surprise the reviewer if chromosomal arm gains/losses were present in the tumors. In the PTEN heterozygous mice, this could be a mechanism to lose the second copy of PTEN.

We thank the reviewer for this suggestion. We did not sequence *Pten* heterozygous mice, as our goal was to examine changes in the Albino *Braf*^{CA/+} tumors. Nonetheless, we did perform coverage-based analysis using the CNVkit algorithm and the Breakdancer algorithm on the sequenced tumors, and report all the variants that were observed (Table 7-12). This was discussed above. We have catalogued all putative CNVs for readers to guide future investigation of individual CNVs that may be deemed to have relevance.

UMAPs can be deceiving, but why do the LNM cells appear to cluster closer to the melanocytes and Schwann cells in figure 2C? Despite the claims by the authors that LNM cells are closer to tumor cells?

We calculated the Euclidean distance between the populations in gene expression space using the top 10 PCs (10 is the number of PC sufficient to cover more than 95% of the variance according to the PC elbow plot). We have updated the Euclidean distance visualization with a heatmap to make it easier to recognize distances between cell populations in gene expression space (Figure S4F). The calculated Euclidean distance between LNM and melanocytes is 70.1, whereas the distance between LNM and Tumor is 56.16, which is consistent with our claim that

LNM cells are more similar to tumor cells than melanocytes. This contention is now supported by Fig S6D where we show that LNM1 is closer in PC space to tumors than LNM2.

In figure 4A Albino BrafCA/+, it appears that the LNM cells cluster to the far left, and this cluster is exclusively skin cells, yet the authors state throughout the manuscript that LNM cells are enriched in tumors. More generally, figure 4 is challenging to follow.

We thank the reviewer for bringing this to our attention. There were cells present in the plot, although we acknowledge they are difficult to see on the figure due to the rare nature of these cells. Moreover, we have found that the histologically superficial nature of LNM cells (they tend to be concentrated near the overlying skin) can sometimes make them difficult to harvest for scRNAseq after the tumor is dissected. To verify that the LNM cells were indeed present, we performed IHC staining with an Aqp1 antibody to verify their presence in black and albino tumors (Fig S7C) and also performed RNAscope to show that some of the LNM cells were *Aqp1*+ and *Sox2*+ (Fig S6E).

References

1. Duffy, D.L. et al. Multiple pigmentation gene polymorphisms account for a substantial proportion of risk of cutaneous malignant melanoma. *J Invest Dermatol* **130**, 520-8 (2010).
2. Ravichandran, S., Funchain, P. & Arbesman, J. Characterizing melanoma in the setting of oculocutaneous albinism: an analysis of the literature. *Arch Dermatol Res* **315**, 2413-2417 (2023).
3. Viros, A. et al. Ultraviolet radiation accelerates BRAF-driven melanomagenesis by targeting TP53. *Nature* **511**, 478-482 (2014).
4. Fan, X., Abbott, T.E., Larson, D. & Chen, K. BreakDancer: Identification of Genomic Structural Variation from Paired-End Read Mapping. *Curr Protoc Bioinformatics* **45**, 15 6 1-11 (2014).
5. Gordeeva, V. et al. Benchmarking germline CNV calling tools from exome sequencing data. *Sci Rep* **11**, 14416 (2021).
6. Dhomen, N. et al. Oncogenic Braf induces melanocyte senescence and melanoma in mice. *Cancer Cell* **15**, 294-303 (2009).

Response to Reviewers:

We thank the three reviewers for their critiques and the opportunity provided to us to respond to them. One reviewer enthusiastically recommended publication (reviewer 2), a second requested clarifications and modification of claims to mitigate concerns (reviewer 1), while a third came up with additional points, not raised in the initial review, addressed below (reviewer 3). Author responses are noted in blue below.

Reviewer #1

Analysis of the comprehensive single cell dataset generated by Xiao et al. supports the emerging concept that non-mutational processes contribute to melanoma formation. While this analysis is timely and valuable, several concerns remain.

1. The mechanism by which the albino allele promotes tumor initiation and triggers LMN expansion is unknown.
2. The source, interchangeability, and relative significance of the LMN1 and LMN2 cell subsets is unclear.

Further clarification and appropriate modification of the authors' claims might mitigate these concerns.

We thank the reviewer for their critique. We acknowledge that we do not yet know how the albino allele promotes tumor initiation/triggers LNM expansion, a topic we leave for future work. We acknowledge that the LNM1 and LNM2 subsets may not necessarily be interchangeable, although we clearly show that both of these populations are marked by the expression of Sox2 and Aqp1. Both Aqp1 and Sox2 are not expressed in melanocytes and are expressed only at a low average level in tumor cells.

Concerns

1. The manuscript concludes that pigment limits the onset of slow-growing melanomas, but the supporting data are based on a single genetic model (BRAF +/- Pten). To make this claim, the authors would need to determine why melanoma formation is enhanced in albino mice or rigorously explore an endless list of other possible explanations. I applaud the authors for making a commendable effort to address a subset of potential mechanisms in a few tumors. However, if identifying the mechanism of enhanced melanoma genesis is beyond the scope of this paper, the authors should avoid claims linking pigmentation or mutation-independent processes to tumor initiation and rapid growth.

Exploring why tumors arise in albino but not black mice is not the main thrust of this paper. We report this phenomenon because it allows us to study a model in which melanomas arise reliably with only one deliberately engineered mutation. We also show that the same sets of cell states arise in *Braf^{CA/+} Pten^{+/-}* models in black mice as in *Braf^{CA/+}* and *Braf^{CA/+} Pten^{+/-}* albinos, but the albino *Braf^{CA/+}* model is particularly useful because it provides the best background on which to look for additional mutations that may have preceded tumorigenesis. We avoid making any direct claims about how pigmentation influences the rate or probability of tumor development, but do acknowledge parallels with clinical and experimental literature on this point.

As for the question of whether an additional mutation-dependent process is required for tumor initiation in the albino *Braf^{CA/+}* model, we do our best to characterize any mutations that were

shared among tumors. There are scenarios we cannot rule out. For example, we cannot rule out the possibility that multiple different mutations might exist that have similar effects in different tumors, or that certain kinds of mutations were missed, nor could we be certain that any mutation that we observe at high frequency in a tumor didn't arise after tumorigenesis and, for reasons having to do either with a fitness effect or neutral drift, came to dominate the tumor cell population.

Despite these caveats, we do report that no established melanoma driver gene, besides *Braf*, is mutated in these tumors, which we believe to be a very important finding. Furthermore, we show that tumor incidence in the albino *Braf^{CA/+}* model behaves as a single-step stochastic process with a half-time of approximately 33 weeks (Fig. 1). As we and other have reported previously^{1, 2, 3}, *Braf^{CA/+}*-transformed melanocytes become growth arrested within about 2 weeks of *Braf^{CA}* activation *in vivo*, implying that whatever stochastic processes is leading to tumor formation takes place in non-dividing cells (a much less likely target for mutation than dividing cells).

2. It is not clear to me that the LMN2 population is a subset of the LMN population shown in Figure 2. This distinction is significant because these cell subsets are used interchangeably throughout the manuscript, and both are hypothesized to promote tumorigenesis. Specifically, the following concerns should be addressed:

a. The relationship between LMN1 and 2 cells is not overly convincing, nor is it clear if the LMN2 subset is truly part of the original LMN subset defined in Figure 2. LMN1 and 2 cells are as different from one another as they are from tumor cells (Euclidean distance = 18.95 and 19.57, respectively). Why do the authors assume that LMN2 cells are a subset of LMN cells? How different are LMN2 cells from the other cell subsets depicted in Figure 2D (i.e. normal melanocytes, Schwann cells)?

To better define the LNM1 and LNM2 subsets, we re-did the clustering, adding additional samples from black and albino tumors so as to better define the clustering space. Given the rarity of LNM1 cells in tumors, increasing the number of such cells improved the ability to characterize similarities and differences among LNM populations, normal melanocytes, and tumor cells. We also updated our measurements of distance, in gene expression space, between populations.

Expression of *Aqp1* and *Sox2* clearly distinguishes both LNM populations and principal tumor cells from all other cells, including normal melanocytes, with LNM1 being characterized by the expression of the highest levels of these genes. Both LNM1 and LNM2 are distinct from principal tumor cells, and about as close to each other in gene expression as they are to principal tumor cells. Because both populations have a "low-pigment, neural and matrix" phenotype we refer to them as LNM1 and LNM2, but that does not imply there is a single "LNM" state of which they are both subsets. The fact that we observe mostly LNM1 cells in primary tumors, a mix of LNM1 and LNM2 in single-round transplants, and mostly LNM2 cells in multi-round-transplanted tumors, is certainly consistent with the possibility that the latter population derives from the former, but we have no direct evidence as to whether that is the case. Still, the fact that both muTRANS and scVelo analyses support the inference that both LNM1 and LNM2 can transition to principal tumor cells lends additional support to the view that LNM1 and LNM2 populations are functionally related.

b. According to the velocity trajectories, LMN2 cells convert to LMN1 cells before becoming a tumor. If this is true, then why do LMN2 cells increase after serial transplantation? Where are the LMN2 cells coming from?

We apologize that the way we previously labeled the RNA velocity figure gave the false impression that LNM2 cells turn into LNM1 cells. In that figure, the LNM1 cluster in the primary tumor sample had been subdivided due to the use of a higher than necessary resolution parameter during unsupervised clustering. The resulting subdivision did not correspond to the LNM1/LNM2 distinction, as nearly all LNM cells in primary tumors are LNM1. We've adjusted the labeling and coloring to make the figure clearer. As for the question of why LNM2 cells increase during rounds of transplantation we can speculate that this reflects selection either for faster growth, or better cell survival, in the host environment. As we do detect a small number of LNM2 cells in primary tumors, such cells may exist from the beginning of tumorigenesis (or earlier), but it is also possible they derive from LNM1 cells or principal tumor cells, as mentioned above. Future work will be required to distinguish among these possibilities.

c. I am not sure how the data in Figure 3D establish that Sox2 and Aqp1 are highly expressed in LNM1 and 2 cells. What is the comparator? Have the authors conducted any statistical analyses?

We now better clarify this point. We compare the expression of Aqp1 and Sox2 in LNM cells as compared to all other cell types in the skin (see Figure 3D, 3E). We detected Aqp1 in 60% of endothelial cells (as previously described)⁴, but also in 95% of LNM1 cells, 34.2% of LNM2 cells and 3% of tumor cells. Sox2, on the other hand, is expressed in >30% of LNM1 and LNM2 cells, and 17% of tumor cells; Sox2 is not expressed in melanocytes or other cell lineages in our dataset (Figure 3D,E).

d. As most LNM1, LNM2, and tumor cells do not express Sox 2 (Figure 3C), how do the authors propose that Sox2 deletion impacts tumor growth? Is it just that homozygous and heterozygous Sox2 knockout mice have fewer melanocytes to transform?

In single cell RNA sequencing, the number of transcripts detected is usually far below the number that is present (scRNAseq merely samples the transcriptome), and it is normal for the majority (often the vast majority) of cells in a population that definitively expresses a gene to display transcript counts of zero for that gene. This is often referred to as the "dropout" problem in scRNAseq, and explains why expression levels are typically displayed as averages over many cells, or using violin plots where counts of zero get de-emphasized. A lack of detected Sox2 in most LNM1 and LNM2 cells does not in any way mean that LNM1 and LNM2 cells don't all express Sox2.

The question of whether loss of Sox2 might be influencing tumorigenesis by acting within melanocytes is not something we address directly, however given that we don't detect Sox2 expression within melanocytes (Fig. 3e) we think it unlikely.

Minor Concerns

1. The homozygous/heterozygous status of the Tyr-CRE(ERT2) allele can influence recombination efficiency. Were all mice in the original study (Fig. 1B) homozygous for this allele? Please provide this information in the manuscript.

All of the mice were in fact heterozygous for the Tyr-CRE(ERT2) allele, and this is now clarified in the methods section of the manuscript.

2. Please label the y-axis in Figure 3D.

We have now included Ridge plots in Figure 3D and 3E where the Y axis is cell type and the X axis is relative expression level. We note the proportion of each cell type expressing Aqp1 or Sox2 in the figure for clarification.

3. The abstract and discussion require revision to eliminate unsupported generalizations regarding the pigmentation and mutational status of fast versus slow-growing melanomas.

References to the pigmentation status and fast versus slow-growing descriptors have been removed from the abstract and discussion. We still include a reference to existing mouse models of tumorigenesis in Figure 6 to highlight the novel observations made in this manuscript.

Reviewer #3 (Remarks to the Author):

I thank the authors for their increased transparency and improved reporting in this revision. Unfortunately, this revealed major issues with the conclusion that tumors are arising via non-genetic events.

In tables S1 and S3, the authors detect *Flna* mutations. In the main text, they claimed “no genes were mutated across all three tumors”, but this is an arbitrary threshold, especially because there is a gene, *Flna*, that is recurrently mutated in two out of three tumors. Of note, the *Flna* mutations have the highest allele frequencies of any coding mutations in these tumors, arguing that they are clonal. Initially, this reviewer was unsure what to make of *Flna* mutations, however, after searching the literature, mutations in this gene are VERY clearly under selection in mouse melanomas. A manuscript by Bowman and colleagues (Life Science Alliance, 2021) from the laboratory of Dr. Christin Burd found mutations in *Flna* in 42% of murine melanomas that developed. While there remain questions as to how *Flna* mutations drive melanoma, given the high frequency of mutations in mouse melanoma, the dN/dS ratio of mutations in mouse melanomas, and now the discovery of recurrent mutations in two independent studies, this gene is clearly a driver gene.

We appreciate the reviewer’s calling attention to *Flna* mutations observed in some tumors. Mutations in *Flna*, a gene on the X-chromosome, are observed at quite low frequency in human melanoma (3.7% of melanomas in the TCGA, all noted as variants of unknown significance in cBioPortal) and have been observed to be under positive selection in a subset of mouse melanomas generated by the Burd⁵ and Marais labs⁶. In one of our tumors the *Flna* mutation is in exon 21, which is also the exon that was most often affected in the mouse melanoma studies of other labs. We observe it in 22% of reads and given an estimated tumor fraction of 55%, that observation is close enough to be consistent with a heterozygous mutation in most or all tumor cells (the tumor arose in a female where heterozygosity is possible). The data are therefore consistent with this mutation being under positive selection in this tumor, although there is no direct way to tell whether it was present at the time of tumor initiation, or rose to high allele frequency afterwards, during tumor progression.

In a separate tumor, from a male, we observed a mutation that produces a 9-nt in-frame deletion of three amino acids in the antepenultimate exon (we refer to this as exon 45, although in some alternative transcripts it is numbered differently). The deletion is seen in 31% of reads. Despite involving *Flna*, we have reason to suspect this mutation may not be directly related to tumorigenesis:

First, the estimated tumor fraction in the sample was 19%, and given the sex of the animal, we would have expected a tumor initiating mutation also to be present at approximately 19%. The fact that we see a level 1.6 times this indicates either that our estimate of tumor frequency was greatly off (even though the estimate came from the same sequencing data), or that the mutation

is actually present in another cell type, e.g., fibroblasts or immune cells (*Flna* is a ubiquitously expressed gene). Second, mutations in this exon were not observed in the mouse melanoma studies mentioned above, and we did not find any when we used our single cell RNAseq datasets to estimate *Flna* mutation frequency in other tumors besides the three that were subjected to whole genome sequencing (described below). Third, according to the three dimensional structure of the protein⁷, the mutation removes three amino acids in a flexible loop between two beta-strands in one of the filamin repeats. Whether this would have functional consequences for that domain is difficult to say without direct structure/function studies. However, we note that no human cancer-associated mutations involving these three amino acids (they are conserved in human) are observed in the COSMIC database, and this portion of the protein is not a hotspot for mutation.

To get a better sense of how often *Flna* mutations arise in our tumors, we also examined the scRNAseq data for seven additional tumors that were not among those analyzed by DNA sequencing. scRNAseq can detect mutations indirectly, although at any given DNA location usually only a handful of cells have informative reads, so this method cannot be used to estimate allele frequency, but it has the advantage that one can limit analysis to particular cell types (e.g. tumor cells). We found non-synonymous *Flna* mutations in 2/7 of these additional tumors, in both cases in exon 21, and confined to the tumor cells. We did not find any mutations involving the C-terminal side of the protein (e.g. in exon 45). Putting this together with the whole-genome sequencing data, we conclude that *Flna* mutations involving exon 21 occur in about 1/3 of our *Braf*-mutant melanomas, which happens to be a frequency very similar to that observed by Burd and Marais in theirs.

Although we now do call attention to mutations in *Flna* as likely playing a role in tumor growth, it should be kept in mind that even mutations observed at “clonal” frequencies did not necessarily arise at the time of tumor initiation. As mentioned above, later mutations can increase to high frequency due to natural selection alone. As tumors expand, the likelihood of a post-initiation mutation increases rapidly (proportional to the number of cells in the tumor), whereas given that *Braf*^{CA/+}-transformed melanocytes become growth arrested within about 2 weeks of *Braf*^{CA} activation *in vivo* (as we and others have reported previously^{1, 2, 3}), a mutational event that triggered tumorigenesis (after, on average, 33 weeks [Fig. 1]) would seem to need to take place in a single, growth-arrested cell (and non-dividing cells accumulate mutations at a rate an order of magnitude lower than replicating cells⁸). None of this alters the fact that *Flna* mutations could be involved in tumor initiation, it simply suggests that alternative mechanisms should be considered.

That is certainly the case for the tumors in which *Flna* mutations were not found (both those analyzed by DNA and RNA sequencing). For example, we did not find mutations in any known driver genes for melanoma initiation, nor in any of the genes found to be recurrently mutated by Burd and Marais in unirradiated melanoma tumors. We also did not observe any other non-synonymous mutations, besides those in *Flna*, in more than one tumor, with the sole exception of *Pira6*, which displayed mutations in two tumors. *Pira6*, however, encodes a gene that, by scRNAseq, is not expressed in melanocytes, tumor cells, or any skin cells, for that matter.

On balance, we believe that our proposal that non-genetic processes may be required for tumor initiation remains worth considering, even for the subset of tumors that bear *Flna* mutations at the time of analysis.

there were other genes and gene families mutated in that study that had mutations reported here in tables S1, S3, and S5. For example, *Vmn* and *Gm* genes were seen in both studies. There is

a strong chance that additional driver mutations, beyond *Flna*, are present but were dismissed due to the unjustifiable requirement that they be present in 100% of tumors.

Mutations called in the *Vmn* genes, which encode olfactory receptors of the vomeronasal organ, are likely artifacts. In the mouse genome there are many hundreds of closely related *Vmn*-receptor genes (319 *Vmn1r* and 228 *Vmn2r* genes), among which are 218 pseudogenes- DNA sequences that contain disabling mutations rendering them unable to produce a fully functional protein⁹, and often created by retrotransposition events (so they are nearly identical to other, true, genes). This creates an enormous mapping challenge for mutation-calling software, particularly when it is based on short reads, so that sequences are sometimes called as mutations in genes when they actually are unmutated reads mapping elsewhere in the genome. Importantly, *Vmn* receptor genes are not generally expressed outside the vomeronasal organ.

Genes noted by the Gm symbol in mice are genes that are not yet otherwise named, which often encode “predicted” proteins or lncRNAs. The Gm symbol does not imply that the genes are in any way related (they are not a gene family). There are several thousand Gm genes annotated in the genome. Overall there are only eight examples of Gm genes in which we observe non synonymous mutations, so it does not appear that mutations in these genes are being detected at a rate any higher than observed in other genes.

While the authors have improved their reporting of point mutations, the reporting of copy number alterations remains subpar. Supplementary tables 7-12 present lists of breakpoints and copy number segments from BreakDancer and CNVkit. A human cannot easily interpret these tables – these outputs were intended to be read by scripts. A better way to present these results would be circos plots (for structural variants) and scatterplots of probe- and segment- level copy number inferences (for copy number changes). As a point of reference, the authors should show copy number data similar to figure 5 of the original CNVkit paper (<https://journals.plos.org/ploscompbiol/article/figure?id=10.1371/journal.pcbi.1004873.g005>).

We thank the reviewer for this suggestion. We have included CIRCOS plots that depict structural variants that were present in skin when compared to spleen and tumors after skin and spleen were subtracted. The number of structural variants in spleen, skin, and tumor samples differed by less 1-3% and there was little overlap between the variants observed in the spleen, skin and tumor (Figure S3A, Tables S9-10, S12-13, S15-16). This suggests that most of the variants may be background, rather than driving the tumor. We included the scatterplots of probe- and segment-level copy number inferences and failed to identify CNVs with overlapping segments in the three tumors where whole genome sequencing was performed (Figure S3B).

The authors do show copy number inferences from inferCNV in supplementary figure 4. In tumor 3, there is clearly a gain on chromosome 6. In tumor 2, there may also be a gain on chromosome 6. These gains are notable because the *Braf* gene is on chromosome 6.

It is important to consider how inferences made by inferCNV originate. The program uses single cell RNAseq data and searches for regions where there is increased expression of contiguous sets of genes, as compared to other cell types. If there are regions of the genome where genes associated with a particular cell state happen to be relatively contiguous, then inferCNV will artifactually call it an amplification. This point is illustrated in the case of tumor macrophages, where CNVs are inferred around the *H2* locus, which is expressed highly in macrophages. Similarly, chromosome 6 happens to contain several of the marker genes of the LNM and tumor cell populations (*Aqp1*, *Col1a2*, *Cav1*, *Cav2*, *Ptn*, *RaRes2*, *Fkbp9*, *Actg2*, *Ybx3*, *Mgp*, *Sox5*, *Kcna1*, *Ptms*). The inferred CNV may have arisen because genes on this chromosome are highly

expressed in LNM cells and tumor cells. Moreover, it is important to point out that even if there were an amplification of Braf on chromosome 6—a possibility the reviewer suggests—this doesn't really change our conclusions. We already know that the Braf mutation is required for tumor initiation, and any such amplification would just provide more evidence that the Braf gene (and not other genes) is involved in initiating tumors.

In conclusion, two tumors have Flna mutations and a third tumor has a broad copy number gain which encompasses Braf. There are other candidate drivers among point mutations, and an improved presentation of copy number data may yet reveal more alterations driving these tumors. The authors describe some interesting non-genetic changes that occur during the early stages of melanoma, but the idea that tumors are arising purely from non-genetic changes does not stand scrutiny.

In this study, we used four different methods to look for mutations in our tumors- Mutect2 to look at SNVs, Breakdancer to look for structural variants, CNVkit to look for copy-number variants, and inferCNV to detect variants that may occur in a small proportion of cells using the single cell data. The reason we examined the data using multiple different pipelines is that each approach makes different kinds of errors (both false positives and negatives). We failed to identify overlaps between data generated from these pipelines, nor did we identify mutations known to be drivers (other than Flna, which is mutated in a minority of mouse tumors, and may or may not be tumor-initiating, as discussed above). We acknowledge in the manuscript that we cannot rule out the possibility that multiple different mutations might have similar effects in different tumors, or that certain kinds of mutations were missed, whether they be SNVs, CNVs, or structural variants, but we still feel that the possibility that initiation involves cell state transitions occurring through non-genetic means needs to be considered.

References

1. Ruiz-Vega R, *et al.* Dynamics of nevus development implicate cell cooperation in the growth arrest of transformed melanocytes. *Elife* **9**, (2020).
2. Dhomen N, *et al.* Oncogenic Braf induces melanocyte senescence and melanoma in mice. *Cancer Cell* **15**, 294-303 (2009).
3. Peeper DS. Oncogene-induced senescence and melanoma: where do we stand? *Pigment Cell Melanoma Res* **24**, 1107-1111 (2011).
4. Verkman AS. Aquaporin water channels and endothelial cell function. *J Anat* **200**, 617-627 (2002).
5. Bowman RL, *et al.* UVB mutagenesis differs in Nras- and Braf-mutant mouse models of melanoma. *Life Sci Alliance* **4**, (2021).
6. Viros A, *et al.* Ultraviolet radiation accelerates BRAF-driven melanomagenesis by targeting TP53. *Nature* **511**, 478-482 (2014).
7. Nakamura F, *et al.* Molecular basis of filamin A-FilGAP interaction and its impairment in congenital disorders associated with filamin A mutations. *PLoS One* **4**, e4928 (2009).

8. Milholland B, Dong X, Zhang L, Hao X, Suh Y, Vijg J. Differences between germline and somatic mutation rates in humans and mice. *Nat Commun* **8**, 15183 (2017).
9. Claes KBM, Rosseel T, De Leeneer K. Dealing with Pseudogenes in Molecular Diagnostics in the Next Generation Sequencing Era. *Methods Mol Biol* **2324**, 363-381 (2021).

Response to Reviewers #3:

We thank the reviewers for their thorough review of the manuscript. All of the issues raised in the reviews could either be clarified typographically or are already clearly stated in the manuscript. A point by point response is provided below (in blue).

Reviewer #1 (Remarks to the Author)

The authors have addressed most of my previous concerns.

We thank the reviewer for acknowledging the effort taken to address their concerns.

The mechanism may be beyond the scope of this publication, but several issues remain regarding the claim that the albino allele leads to rapid tumor growth.

The copy number of Tyr:CreERT(2) may influence the rate of tumorigenesis. The authors state that all study mice were heterozygous for Tyr:CreERT(2). However, the provided genotyping primers only amplify Cre, and the breeding scheme does not preclude the birth of homozygous animals. The authors should use a primer set specific to the Tyr:CreERT(2) insertion site (see: Aktary et al.; PMID: 31679174) to confirm the zygosity of the experimental mice. Once confirmed, these data should be included in the Results section, and both breeding schemes shown in Supplementary Figure 1.

We did use the PCR primers suggested to verify the heterozygosity of the animals, but the reference was not included and the primer sequences were incorrect, and we thank the reviewer for catching this. We have changed the sentence on line 421 to state: “All mice used in the study were heterozygous for the *Tyr::CreER* allele, which was verified by primers that could distinguish the wild type and mutant alleles.” While it is possible to include some representative gels (such as shown below), we feel it would be simpler to address by updating the reference and primer sequence (sequence has been updated in the key resources table).

Figure 1: Representative genotyping gel using the Bos-Cre primers. PCR primers were used to identify the *Pten*^{fl} allele (first lane) or to distinguish the wild type (WT) from the CreERT2 transgenic allele (Tg) using the Bos-Cre primers. Only those mice that were heterozygous for the transgene were used in this study.

Were the pups from the second cross in Supplementary Figure 1C (left panel, right side of the cross) separated before tamoxifen treatment to ensure those used for subsequent

breeding were not exposed to tamoxifen? Although these animals would constitute only a small portion of the study, it is important to examine whether prior tamoxifen exposure did enhance tumorigenesis in the offspring.

These mice were separated before tamoxifen treatment to ensure those used for subsequent breeding were not exposed to tamoxifen. The figure legend text has been updated to reflect this.

There is insufficient evidence to show that the loss of pigment, rather than Tyrosinase function, leads to changes in melanoma development. Therefore, it is suggested that the authors:

- *Remove the claim about pigmentation in the following sentence (lines 101-103): “Nonetheless, it is clear that pigmentation status and Pten gene dosage both influence the probability of melanoma initiation in *Braf*^{CA/+} and *Braf*^{CA/+} *Pten*^{D/+} mice.”*

We have changed this sentence to the following to satisfy the reviewer: Nonetheless, it is clear that tyrosinase function and *Pten* gene dosage both influence the probability of melanoma initiation in *Braf*^{CA/+} and *Braf*^{CA/+} *Pten*^{Δ/+} mice.

- *Replace “albino” with “Tyrosinase-deficient” in the statement on lines 371-373.*

We have made this change.

- *Consider revising the model in Figure 6, replacing “albino” with “Tyr C-2J homozygous.”*

We have edited this model appropriately.

Reviewer #3 (Remarks to the Author)

*I believe the in-frame indel in *Flna* of tumor 2 is a driver mutation. In argument against this position, the authors begin by mentioning that *Flna* mutations are rare in human melanoma, but this is irrelevant because they are drivers in mouse melanoma. Next, the authors note that this *Flna* mutation has an allele frequency that is 1.6X the inferred tumor cell content. The high allele frequency of the in-frame indel could have easily arisen from random over sampling of the mutant allele in the sequencing data, and/or the authors estimate of tumor cellularity could be too low. The authors next point out that the in-frame indel has not yet been observed in other mouse melanomas. While the exact (or similar) indel mutation in *Flna* might not have been observed in another tumor (as of yet), it is extremely common for cancer genes to have minor hotspots. As an example, one can look at the distribution of *BRAF* mutations in human melanomas – V600E is most common, but there are plenty of*

other hotspots, mutated at less than 5% frequency. The mutation burden of the mouse tumors in this study is low and considering this, it is difficult to fathom how a passenger mutation arose in a driver gene. Taken together, this reviewer believes that Flna driver mutations are present in 2 out of the 3 tumors that were deeply characterized by the authors, and apparently they are in additional tumors with single-cell RNA sequencing data, which is telling because scRNA-seq is not a sensitive method to detect mutations.

We appreciate the reviewers concern but would like to clarify several points here. First, we would like to clarify that the distinction we are making here is not between “driver” and “passenger”, but between tumor initiating mutations and those that helped drive tumor progression. Because tumors cannot be sampled at the time of initiation, one cannot clearly distinguish a mutation required for initiation from one that was selected for during tumor progression. We specifically allude to this fact in the discussion where we state we cannot “be certain that any mutation that we observe at high frequency in a tumor didn’t arise after tumorigenesis and, for reasons having to do either with a fitness effect or neutral drift, came to dominate the tumor cell population” (lines 386-388).

To clarify, we note that only the exon 21 mutation—not the indel the reviewer is referring to—is also observed in the scRNAseq data in tumors. The reviewer’s statement that “scRNA-seq is not a sensitive method to detect mutations” is actually misleading in this context. We are looking here for initiating mutations which should by definition be present in either 50% or 100% of the DNA from tumor cells. The use of scRNAseq allows one to limit analysis just to bona fide tumor cells—one does not have to correct for the presence of non-tumor DNA—so as long as one has enough tumor cells so that there are more than a handful of reads covering each base, it is quite difficult to miss an initiating mutation. This is why, in Table S8, we were able to confidently call those samples in which FLNA mutations were detected in exon 21. When we look at further downstream exons (such as exon 45 which are actually sequenced more deeply than exon 21, a phenomena typical of scRNAseq that uses 3-prime biased libraries), we can have even greater confidence in any mutations identified. Thus, we have more sensitivity to detect the exon 45 mutation as compared to the exon 21 mutation. Thus, the fact that this mutation was not identified in our single cell data indicates that this is an uncommon event.

Nonetheless, we would like to point out that we have acknowledged such caveats, stated on lines 384-386: “There are scenarios we cannot rule out. For example, we cannot rule out the possibility that multiple different mutations might exist that have similar effects in different tumors, or that certain kinds of mutations were missed”. These caveats acknowledge the reviewer’s point and address their concern.

While this reviewer does not recognize an obvious driver mutation in the third tumor or additional driver mutations beyond the Flna mutation in the first two tumors, there is quite a lot of noise in every genetic result in the manuscript.

We would like to point out that when the reviewer is talking about “noise”, they may be conflating issues of false negativity and false positivity. In order to make the chance of our missing mutations and structural changes as low as possible (i.e., to reduce false negativity) we deliberately use parameters, conditions and methods that are associated with a high degree of false positivity. It is thus inaccurate to characterize this as “a lot of noise in every genetic result in the manuscript”, but on the contrary, it is an indication of our efforts to be as scrupulous as possible in our analysis and conclusions. An observation that, after having gone to such lengths, no candidate driver mutations appear in a tumor, even at such a high rate of false positivity, cannot be so easily dismissed.

To address the reviewer’s continuing concern, we go out of our way in the manuscript to make clear the limitations of each of the methods we use: “...each method to identify mutations has caveats: short-read sequencing misses many structural variants; the power of bulk methods is diminished by the presence of non-tumor cells in tumor samples; inference of CNVs from gene expression data can produce false positive results when groups of nearby genes exhibit cell-type specific expression; and methods designed to identify variants with high confidence (i.e. with few false positives) are not necessarily well-powered statistically to rule out the presence of variants (i.e. display few false negatives”

For example, the authors point out that their point mutations are potentially full of artifacts in Vmn and Gm genes.

We would like to point out that this may again be conflating false positivity and false negativity. Our primary goal, to address the reviewer’s concern, is to avoid false negatives. We could have skipped over genes like vomeronasal receptor genes and certain Gm genes based on the difficulties in mapping short read sequences to these genes, but we did not because we did not want any stone to be unturned. The price of that is we expect false positives (“artifacts”). Importantly, none of these genes are associated with tumorigenesis, nor are the Vmn receptor genes even expressed in melanocytic lineage cells.

Overall, I do agree when the reviewer say “our proposal that non-genetic processes may be required for tumor initiation remains worth considering”. The authors insinuate that non-genetic processes may be sufficient for tumor initiation – the repetitive claims that they failed to find driver mutations in a majority of tumors certainly lends itself to this interpretation. My take is that genetic processes are occurring alongside non-genetic processes to drive the evolution of the melanomas in this study. I am not convinced that these tumors are arising in the absence of genetic drivers.

We appreciate that the reviewer does not dispute the main contention in our manuscript: “our proposal that non-genetic processes may be required for tumor initiation remains worth considering.” We have adequately noted their concern, as noted above, and explicitly stated that we cannot rule out that there were tumor initiating mutations that were missed. Thus, we have presented the data in a fair and balanced way.

Regardless of whether one believes that tumors could arise solely in the absence of additional genetic drivers, the findings are highly relevant which we highlight in the discussion (lines 399-400): “In contrast, while new genetic variants are not conserved between the tumors, the cell state and cell state transition proposed to initiate melanoma tumors was seen in every case studied here”. The implication of this statement is that the cell state identified here may provide a diagnostically and therapeutically important source of cellular heterogeneity, whether the state was induced by multiple different mutations or not.

References

1. Aktary Z, Corvelo A, Estrin C, Larue L. Sequencing two Tyr::CreER(T2) transgenic mouse lines. *Pigment Cell Melanoma Res* **33**, 426-434 (2020).